



# Intercomparison of Sentinel-5P TROPOMI cloud products for tropospheric trace gas retrievals

Miriam Latsch[1], Andreas Richter[1], Henk Eskes[2], Maarten Sneep[2], Ping Wang[2], Pepijn Veefkind[2], Ronny Lutz[3], Diego Loyola[3], Athina Argyrouli[3,7], Pieter Valks[3], Thomas Wagner[4], Holger Sihler[4], Michel van Roozendael[5], Nicolas Theys[5], Huan Yu[5], Richard Siddans[6], and John P. Burrows[1]

[1]Institute of Environmental Physics, University of Bremen (IUP-UB), Otto-Hahn-Allee 1, 28359 Bremen, Germany
[2]KNMI, Royal Netherlands Meteorological Institute, De Bilt, the Netherlands
[3]Remote Sensing Technology Institute, German Aerospace Centre (DLR), Wessling, Germany
[4]Max Planck Institute for Chemistry (MPIC), Mainz, Germany
[5]Royal Belgian Institute for Space Aeronomy (BIRA-IASB), Brussels, Belgium
[6]Science and Technology Facilities Council, Rutherford Appleton Laboratory (RAL), Chilton, UK
[7]Technical University of Munich, Department of Civil, Geo and Environmental Engineering, Munich, Germany

*Correspondence to*: Miriam Latsch (mlatsch@iup.physik.uni-bremen.de)

**Abstract.** Clouds have a strong impact on satellite measurements of tropospheric trace gases in the ultraviolet, visible, and near-infrared spectral ranges from space. Therefore, trace gas retrievals rely on information on cloud fraction, cloud albedo, and cloud height from cloud products. In this study, the cloud parameters from different cloud retrieval algorithms for Sentinel-5 Precursor (S5P) TROPOMI are compared: the OCRA a priori cloud fraction, the ROCINN CAL cloud fraction and cloud top and base height, the ROCINN CRB cloud fraction and cloud height, the FRESCO cloud fraction, the interpolated FRESCO cloud height from the TROPOMI $NO_2$ product, the cloud fraction from the $NO_2$ fitting window, the O2-O2 cloud fraction and cloud height, the MICRU cloud fraction, and the VIIRS cloud fraction. Two different versions of the TROPOMI cloud products OCRA/ROCINN, FRESCO, and the TROPOMI $NO_2$ product are included in the comparisons (processor version 1.x and 2.x). Overall, the cloud parameters retrieved by the different algorithms show qualitative consistency in version 1.x and good agreement in version 2.x with the exception of the VIIRS cloud fraction, which cannot be directly compared to the other data. Differences between the cloud retrievals are found especially for small cloud heights with a cloud fraction threshold of 0.2, i.e., clouds that are particularly relevant for tropospheric trace gas retrievals. The cloud fractions of the different version 2 cloud products primarily differ over snow- and ice-covered pixels and scenes with sun glint, for which only MICRU includes an explicit treatment. All cloud parameters show some systematic problems related to the across-track dependence, where larger values are found at the edges of the satellite view. The consistency between the cloud parameters from different algorithms depends strongly on how the data is filtered for the comparison, e.g., what quality value is used or whether snow- and ice-covered pixels are excluded from the analysis. In summary, clear differences were found between the results of various algorithms, but these differences are reduced in the most recent versions of the cloud data.



# 1 Introduction

Monitoring the global distribution of atmospheric constituents over a long period is essential to assess changes in atmospheric composition and the resulting consequences, such as pollution and climate change. One efficient way to perform such measurements is by using absorption spectrometry from satellite platforms. The first nadir-viewing instrument to measure with high spectral resolution in the ultraviolet/visible/near-infrared (UV/VIS/NIR) spectral range, GOME aboard ERS-2 (Burrows et al., 1999 and references therein), was launched in April 1995 to detect multiple species including ozone ($O_3$), nitrogen dioxide ($NO_2$), bromine monoxide (BrO), sulphur dioxide ($SO_2$), and formaldehyde (HCHO). During 2002-2012, SCIAMACHY onboard ENVISAT (e.g., Burrows et al., 1995; Bovensmann et al., 1999) enabled the observation of additional species such as carbon monoxide (CO), carbon dioxide ($CO_2$), and methane ($CH_4$), through the integrated short-wave infrared (SWIR) channels. OMI launched in 2004 on the Aura platform (Levelt et al., 2018) and the three GOME-2 instruments (Munro et al., 2016) launched in 2006 aboard Metop-A, in 2012 on Metop-B, and in 2018 on Metop-C expand the atmospheric composition data set. The ESA Sentinel-5 Precursor (S5P) with the TROPOspheric Monitoring Instrument (TROPOMI) on board (Veefkind et al., 2012) was launched in October 2017 as a preparatory mission to bridge the gap between the existing satellites and the planned Sentinel-4 and Sentinel-5 missions. TROPOMI is a space-borne nadir-viewing hyperspectral imaging spectrometer covering the UV, VIS, NIR, and SWIR spectral regions. TROPOMI monitors atmospheric trace gases daily and globally with a high spatial resolution of 5.5 x 3.5 $km^2$ (5.5 x 7 $km^2$ before 6 August 2019) (Eskes et al., 2021).

Satellite measurements of atmospheric composition are affected by clouds as they shield the underlying atmosphere from the satellite's view, reducing the sensitivity to the lower atmosphere. At the same time, clouds increase the sensitivity for absorbers above or inside the cloud through their higher albedo and multiple scattering within the cloud. As a result, it is challenging to retrieve the amounts of trace gases when clouds are in the field of view of the instrument due to increased reflection of solar radiation and enhanced light paths (e.g., Martin et al., 2002; Richter and Burrows, 2002; Boersma et al., 2004). Consequently, cloud retrieval algorithms have been developed and implemented in trace gas processing to account for the cloud impact when observing atmospheric gases with spectrometers from space (e.g., Koelemeijer et al., 2001; Loyola et al., 2004; Kokhanovsky et al., 2006; Stammes et al., 2008; Lutz et al., 2016). In general, cloud retrieval algorithms use the Independent Pixel Approximation (IPA), which defines the scene as a linear combination of a cloudy and a clear sub-scene. In the cloud-free sub-scene, part of the solar light reaches the surface and is reflected back to the satellite. In the cloudy part of the scene, the solar light is scattered and reflected by the cloud, which affects the amount of absorption by atmospheric trace gases.

For TROPOMI, several cloud products have been developed that use different physical processes as approaches to retrieve cloud parameters such as cloud fraction, cloud height, and cloud optical thickness. The Optical Cloud Recognition Algorithm (OCRA) and the Retrieval Of Cloud Information using Neural Networks (ROCINN) are the operational cloud product for TROPOMI (Loyola et al., 2018; Loyola et al., 2021; see 2.1.1). The colour or whiteness approach is implemented in OCRA to retrieve a radiometric cloud fraction. OCRA applies background maps in a normalized red-green-blue (RGB) colour space, where an optically thick cloud is assumed to be white. In ROCINN the $O_2$ absorption band is used in the range 756-771 nm to



provide cloud height information as well as cloud optical thickness and cloud albedo. The Fast Retrieval Scheme for Clouds from the Oxygen A-band (FRESCO) uses the $O_2$ A-band and the brightness approach for the NIR region (Wang et al., 2008; see 2.1.2). The brightness approach, where a cloud-free background is defined as dark compared to bright clouds, is also implemented in the TROPOMI $NO_2$ product for the UV/VIS region (Van Geffen et al., 2021; see 2.1.2), in the Mainz Iterative Cloud Retrieval Utilities (MICRU) for the UV/VIS/NIR region (Sihler et al., 2021; see 2.1.4), and in the Visible Infrared

Imaging Radiometer Suite (VIIRS) for the VIS/IR/SWIR region (Siddans, 2016; see 2.1.5). $O_2$-$O_2$ absorption in the spectral range between 460 nm and 490 nm is used for the TROPOMI O2-O2 product (Acarreta et al.,2004; Veefkind et al., 2016; see 2.1.3). This approach was initially developed for OMI because the spectral range of the $O_2$ A-band is not covered by this instrument. In general, cloud retrieval algorithms use cloud models where clouds are assumed to be, e.g., reflecting surfaces as in OCRA/ROCINN CRB (Clouds as Reflecting Boundaries), FRESCO, MICRU, and O2-O2, or homogeneous layers as in

OCRA/ROCINN CAL (Clouds As scattering Layers). The cloud products differ in how the cloud albedo is determined, either it is fitted to the scene as in OCRA/ROCINN or assumed to have a fixed value of 0.8 as in FRESCO, O2-O2, and MICRU.

Sihler et al. (2021) present the MICRU algorithm in more detail, comparing MICRU and different versions of OCRA and FRESCO using GOME-2 data. They show that MICRU is able to accurately determine small cloud fractions over a wide spectral range with less dependence on sun glint. Compernolle et al. (2021) performed a comprehensive validation of the

80 OCRA/ROCINN CAL and CRB models as well as the FRESCO cloud product using cloud data from the NPP-VIIRS, OMI, and MODIS satellites and ground-based data from CLOUDNET. They present a new method for comparing the OCRA/ROCINN CRB cloud fraction and the FRESCO cloud fraction, converting the former to a scaled cloud fraction with a cloud albedo of 0.8, which is assumed in the FRESCO and O2-O2 products. This procedure is used in the comparisons in this paper (see 2.2). Compernolle et al. (2021) found a pronounced west-east bias in the version 1 OCRA/ROCINN cloud product

and unrealistic cloud heights equal to the surface altitude at low cloud fractions in the version 1 FRESCO product.

In this study, the operational TROPOMI cloud product (consisting of OCRA a priori, ROCINN CRB, and ROCINN CAL), the FRESCO cloud product, the cloud fraction from the $NO_2$ fitting window, the O2-O2 cloud product, the VIIRS cloud fraction, and the cloud fraction from MICRU are compared with respect to different regions (Europe, Africa, and China) and four test days in different seasons, i.e., a summer day (30 June 2018), a winter day (5 January 2019), a spring day (4 April

2019), and a fall day (20 September 2019). The OCRA/ROCINN CLOUD, FRESCO, $NO_2$, and VIIRS products were updated from version 1 to version 2 in summer 2020, and both versions are included in the comparison. This study is limited to the four test days, as data from both versions are only available for specific days. The goal of this paper is to summarize and compare the existing cloud retrieval algorithms for TROPOMI, to discuss their differences and to document the changes between the versions. The focus is on parameters needed for the application of the cloud products for trace gas retrievals, not

the retrieval of cloud properties themselves.

The manuscript is organized in the following way: The different TROPOMI cloud products, their properties, and the data preparation are described in Section 2. In Section 3, the cloud products are statistically compared first with respect to the



version change (3.1), and second regarding the version 2 cloud fractions (3.2.1) over snow- and ice-covered areas (3.2.2) and with sun glint (3.2.3), cloud heights (3.2.4), and across-track dependencies (3.2.5). Finally, conclusions are given in Section 4.

## 2 Methods

In this study, we compare different cloud products based on TROPOMI data of version 1 and version 2. This section presents first the input data and the different cloud retrieval algorithms (2.1). In Section 2.2, the data preparation for the comparison of the different TROPOMI cloud products is described.

### 2.1 Cloud retrieval algorithms

For TROPOMI, different cloud retrieval algorithms have been developed to retrieve cloud parameters from the UV, VIS, and NIR spectral regions. As the cloud products use different approaches, the retrieved cloud fractions, cloud albedo, and cloud heights differ. An overview of the cloud products used in this paper is shown in Table 1.

**Table 1: Overview of the cloud products with abbreviations for the version 1 (v1) and version 2 (v2) cloud fraction and cloud height parameters included in this study, their approaches and the spectral ranges. MICRU and O2-O2 have only one version.**

| cloud product | cloud fraction | cloud height | approach | spectral range |
|---|---|---|---|---|
| OCRA a priori | cf_apriori v1 / v2 | - | colour (whiteness) | 350-495 nm |
| ROCINN CAL | cf_cal v1 / v2 | ch_cal v1 / v2 | $O_2$ absorption | 758-771 nm |
| ROCINN CRB | cf_crb v1 / v2 | ch_crb v1 / v2 | $O_2$ absorption | 758-771 nm |
| FRESCO | cf_fresco v1 / v2 | - | brightness, $O_2$ absorption | 758-766 nm |
| NO$_2$ | cf_fit v1 / v2 | ch_fresco* v1 / v2 | brightness | 440 nm |
| O2-O2 | cf_o2o2 | ch_o2o2 | O2-O2 absorption | 460-490 nm |
| MICRU | cf_micru | - | brightness | 375-757 nm |
| VIIRS | cf_viirs v1 (VCM) / v2 (ECM) | - | brightness | 412-12000 nm |

### 2.1.1 CLOUD OCRA/ROCINN

The operational TROPOMI CLOUD product generated from the Universal Processor for UV/VIS Atmospheric Spectrometers (UPAS) was developed by the German Aerospace Centre (DLR) as a two-step algorithm. First, OCRA for TROPOMI, an algorithm for cloud detection by optical sensors, is applied to TROPOMI measurements in the UV/VIS spectral region to retrieve the cloud fraction a priori (Loyola et al., 2018, Loyola et al., 2021). Using the colour-space approach, the UV/VIS reflectances of the observed scene are translated to colours to obtain the radiometric cloud fraction. In UPAS 1.x, which was operational until July 2020, the clear-sky reflectance and the across-track dependency correction are based on OMI data with a spatial resolution of 0.2° x 0.4°. In UPAS 2.1.3, operational between July 2020 and July 2021, the clear-sky background map and the across-track dependency correction are based on one year of TROPOMI data with a spatial resolution of 0.2° x 0.2°





and since UPAS 2.2.1, those are based on three years of TROPOMI data. In addition, an adapted scaling is included to improve the range of very low and very high cloud fractions.

Second, the OCRA a priori cloud fraction and NIR TROPOMI measurements are taken as input to a machine learning algorithm, ROCINN, to retrieve the cloud-top height, the cloud optical thickness, and the cloud albedo from reflectivity measurements in and around the $O_2$ A-band between 758 and 771 nm. Two cloud models are implemented in ROCINN: the Clouds As scattering Layers (CAL) model and the Clouds as Reflecting Boundaries (CRB) model. ROCINN CAL treats clouds as homogeneous layers of scattering liquid water particles to retrieve cloud fraction, cloud top height, and cloud optical thickness. The cloud base height from ROCINN CAL is not a retrieved quantity; instead, the cloud is assumed to have a constant geometrical thickness of 1 km. In ROCINN CRB, clouds are Lambertian equivalent reflectors with cloud fraction, cloud height, and cloud albedo as output. Cloud fractions that are smaller than 0.05 in OCRA a priori are set to zero in the ROCINN CAL and CRB cloud fractions, and the ROCINN retrieval is not triggered under these "clear-sky" conditions. ROCINN used a MEdium Resolution Imaging Spectrometer (MERIS) based surface albedo climatology in UPAS 1.x. Starting with UPAS 2.1.3, the surface albedo climatology is replaced by an actual surface albedo retrieval of geometry-dependent effective Lambertian equivalent reflectivity (GE_LER) using the TROPOMI data, and the surface albedo map is dynamically updated every day with the global gapless geometry-dependent LER (G3_LER) (Loyola et al., 2021) if a scene is indicated as clear-sky. In addition, cloud phase flags and effective scene parameters such as effective scene height and effective scene albedo are added, and the co-registration between the UV/VIS (BD3) and NIR (BD6) bands is improved.

This study includes the UPAS 1.1.7 data as version 1 and the UPAS 2.1.3 data as version 2. It should be noted that a very simple initial approach with limited usability is used for the quality value in UPAS 1.x, while significant improvements in the determination of the quality values have been made in UPAS 2.1.3 compared to version 1.x. In both versions, no quality filtering is applied for the OCRA a priori cloud fraction.

The operational OCRA/ROCINN products are used for the cloud correction of the following operational TROPOMI products: total ozone (Spurr et al., 2021), tropospheric ozone (Heue et al., 2018), $SO_2$ (Theys et al., 2017) and HCHO (De Smedt et al., 2018).

### 2.1.2 FRESCO and TROPOMI NO$_2$ product

The FRESCO algorithm, developed by the Royal Netherlands Meteorological Institute (KNMI), models the effective cloud fraction and cloud pressure (height) using the $O_2$ A-band centred at 760 nm (Koelemeijer et al., 2001; Wang et al., 2008). The cloud parameters are retrieved from top-of-atmosphere reflectances in three 1 nm wide wavelength windows at 758-759 nm (no absorption), 760-761 nm (strong absorption), and 765-766 nm (moderate absorption). Measurements from the NIR spectrum, where land is characterised by a high albedo in contrast to dark water, make FRESCO susceptible to uncertainties in the surface reflectance, such as distinct coastlines and a land-water-contrast. FRESCO uses a Lambertian cloud model, where the cloud is assumed to be a Lambertian reflector with a fixed albedo of 0.8. The processor version of FRESCO defined as version 1 is 1.3.x, version 2 refers to FRESCO processor version 2.1.0, which uses new look-up tables for the surface albedo



and degradation-corrected irradiances. The FRESCO implementation in processor version 1.4, operational since December 2020, and later processor versions 2.x have adopted a different, wider wavelength window in the oxygen A-band (working title: FRESCO-wide). This generally leads to lower cloud pressures, correcting a high bias observed in versions 1.2 and 1.3, and to significant increases in $NO_2$ in better agreement with OMI $NO_2$ retrievals (Van Geffen et al., 2022).

The cloud retrieval algorithm of the TROPOMI $NO_2$ product has been developed due to the misalignment between the TROPOMI ground pixel view of the VIS and NIR bands (Van Geffen et al., 2021). The effective cloud fraction from the TROPOMI $NO_2$ product is retrieved from the $NO_2$ fitting window in the UV/VIS spectral region at 440 nm. The cloud height of the TROPOMI $NO_2$ product is derived from the FRESCO cloud pressure of the TROPOMI FRESCO product, taking into account the difference in the footprint of the UV/VIS and NIR detectors. The FRESCO cloud height of the TROPOMI $NO_2$ product is very similar to that of the TROPOMI FRESCO product. The only difference is that the FRESCO cloud height is not corrected for the misalignment between the UV/VIS and NIR channels of TROPOMI. Consequently, only the cloud height from the TROPOMI $NO_2$ product is included in the comparisons in this paper, and the FRESCO cloud height is not dealt with explicitly. For the TROPOMI $NO_2$ product, the processor versions 1.2.2 and 1.3.x are used as version 1 and the processor version 2.1.0 is used as version 2.

### 2.1.3 O2-O2

The O2-O2 algorithm has been developed by KNMI and was initially developed for OMI, because this instrument does not cover the spectral range of the $O_2$ A-band at 760 nm (Acarreta et al., 2004; Veefkind et al., 2016). The algorithm uses OMI/TROPOMI measurements from the O2-O2 ($O_4$) absorption window at 477 nm to retrieve the effective cloud fraction and the cloud height using a similar cloud model as the one used in FRESCO. However, it is more sensitive to clouds at lower altitudes and to aerosols because it uses O2-O2 collision-induced absorption. As in FRESCO, a fixed cloud albedo of 0.8 is assumed. The retrieved cloud height is expected to be the mid-level of the cloud rather than the cloud top height (Sneep et al., 2008). Since TROPOMI processor version 2.2.0 (van Geffen et al., 2022), the O2-O2 cloud product is included in the TROPOMI $NO_2$ retrieval files, but the $NO_2$ retrieval does currently only use the FRESCO cloud information. For the O2-O2 cloud product, the processor version 2.2.0 is used in this study.

### 2.1.4 MICRU

The MICRU algorithm was designed by the Max-Planck-Institute for Chemistry (MPIC) to retrieve the effective cloud fraction at different spectral bands using UV/VIS/NIR TROPOMI measurements (Sihler et al., 2021). MICRU is optimized for low cloud fractions smaller than 0.2. It uses a viewing direction dependent empirical background map of surface reflectivity and differentiates between land and ocean. MICRU only computes effective cloud fractions and no other cloud parameters.



### 2.1.5 VIIRS

The VIIRS instrument is aboard the Suomi National Polar-orbiting Partnership (NPP) satellite platform launched in 2011. The S5P-NPP cloud product has been developed by the Rutherford Appleton Laboratory (RAL) to retrieve a 4-level cloud mask with cloud probability for VIIRS pixels within an S5P scene (Siddans, 2021). VIIRS VIS and infrared (IR) imagery and

radiometric measurements are used as input to obtain a geometric cloud fraction, which is based on the cloud mask and is mainly independent of the cloud optical properties. It should be emphasized that the effective cloud fractions retrieved from the cloud products mentioned above strongly depend on the cloud optical thickness, i.e., for optically thick clouds, the effective cloud fraction may be close to the geometric one, but for optically thin clouds it might be much below. Therefore, the geometric VIIRS cloud fraction is expected to have the largest differences from the other cloud fractions. In this study, the cloud fraction

calculated from the VIIRS Cloud Mask (VCM) included in the TROPOMI L2_NP_BD3 product file of processor version 1.0.2 is defined as version 1 (see Section 2.2 for details of the calculation). The cloud fraction from the VIIRS Enterprise Cloud Mask (ECM) is defined as version 2 and is directly taken from the TROPOMI CLOUD product file of processor version 2.1.3.

### 2.2 Data preparation

Throughout this study, data with a quality parameter (qa value) larger than or equal to 0.5 is used. The qa value ranges between

0 and 1 and is not a mathematical parameter, but an artificial value used to decide whether the pixels are of good quality (1) or whether the measurement is affected by processing errors and warnings, in which case the qa value is reduced from 1. There are no common rules on how the qa value should be calculated and what input information should be included (e.g., sun glint, snow/ice, low cloud fractions, aerosol pollution, extreme viewing geometries, algorithm-specific retrieval diagnostics). Therefore, the qa values in the different products (ROCINN CRB, ROCINN CAL, FRESCO, and TROPOMI $NO_2$ product)

are not directly comparable, and this non-standardized qa value calculation across all products has a large impact on the comparison of the cloud data in this study as will be seen later.

Some parameters from the set of TROPOMI cloud products need to be further processed to make the results from the different cloud retrieval algorithms more comparable.

*ROCINN Cloud Albedo Scaling*

As recommended by Compernolle et al. (2021), the ROCINN CRB cloud fraction (CF) is converted to a scaled cloud fraction (sCF) with a fixed cloud albedo (CA) at 0.8. This provides a better comparison to cloud products that assume a fixed cloud albedo (0.8), such as FRESCO or O2-O2. The CA is assigned a fill value when CF = 0, thus the scaling is done as follows:

$$sCF = \frac{CF * CA}{0.8}, \text{ if } CF > 0 \tag{1}$$

$sCF = 0, \text{ if } CF = 0$



*Pressure to Height Conversion*

The parameter "cloud_pressure_crb" (CP in Pascal) from the TROPOMI NO$_2$ product needs to be converted to cloud height (CH in meters). This parameter is derived from the FRESCO cloud pressure, already considering the difference in the footprint of the NIR and UV/VIS detectors. The following conversion formula is used:

$$CH = \frac{T_0}{L} * \left( \frac{CP}{p_{surface}}^{\frac{L*R_s}{g}} - 1 \right) \qquad (2)$$

where $L = -0.0065$ K m$^{-1}$ is the constant tropospheric temperature lapse rate, $p_{surface} = 101325$ Pa is the surface pressure, and $g = 9.81$ m s$^{-2}$ is the gravitational acceleration constant. The surface temperature $T_0$ is assumed to be 300 K. $R_s = \frac{R}{MW_{air}}$ is the specific gas constant for air with the universal gas constant $R = 8.3144621$ J mol$^{-1}$ K$^{-1}$ and the molar weight of air $MW_{air} = 0.0289644$ kg mol$^{-1}$. The FRESCO and NO$_2$ cloud heights are the same quantity, spatially interpolated for different spectral bands. FRESCO is retrieved in the NIR region, and the FRESCO cloud height in the TROPOMI NO$_2$ product is interpolated to the TROPOMI pixel coordinates of the UV/VIS region. As explained in Section 2.1.2, only the NO$_2$ cloud height is considered in the comparisons in this study.

*Other Data Operations*

FRESCO and O2-O2 cloud fractions reach values up to 1.5, which is physically impossible. This is due to the assumption of a fixed albedo of 0.8. Thus, values larger than 1.0 are forced to 1.0 to ensure a consistent comparison and are expected to accumulate in the scatter plots.

The dimension "ground_pixel" of the FRESCO parameters has only 448 pixels compared to the other products, which have 450 values. This is a consequence of the different spatial coverage (i.e., footprints) of the NIR and UV/VIS channels of TROPOMI. Therefore, the "ground_pixel" dimension of the FRESCO parameters is filled with two additional entries at the beginning of every scanline to ensure comparability. No attempt was made to map the FRESCO cloud fraction to the footprint of the UV/VIS channel. Therefore, some scatter is expected in the comparisons involving FRESCO cloud fraction data.

The MICRU cloud fraction derived at 440 nm is included in the comparisons because this wavelength corresponds to a point within the NO$_2$ fitting window (Sihler et al., 2021).

The ratio of the sum of pixels in the VIIRS class of interest ("vcm_confidently_cloudy") and the total number of all pixels, in each case for the nominal field-of-view, is calculated to determine the geometric cloud fraction of the VIIRS measurement (Siddans, 2021).

In this study, the results are presented in scatter plots, where the colour bar indicates what percentage of the available values is accumulated in the cloud fraction or cloud height values in the dots. In addition, the cloud products are compared in terms of differences on maps. For that purpose, the TROPOMI measurement pixels of all orbits are rasterized to a 0.03° x 0.03° grid. Thus, the differences represent averaged values of the measurements. This is important at higher latitudes where orbits overlap and measurements from multiple orbits are available per grid pixel.





## 3 Results and Discussion

Due to limited data availability of TROPOMI version 2 data, the comparisons of the different cloud products are restricted to four test days from different seasons: 30 June 2019 (summer day), 5 January 2020 (winter day), 4 April 2020 (spring day), and 20 September 2020 (fall day). In addition, the cloud products are investigated for three defined regions, namely Europe, China, and Africa, whose spatial definitions can be seen in Figure 1.

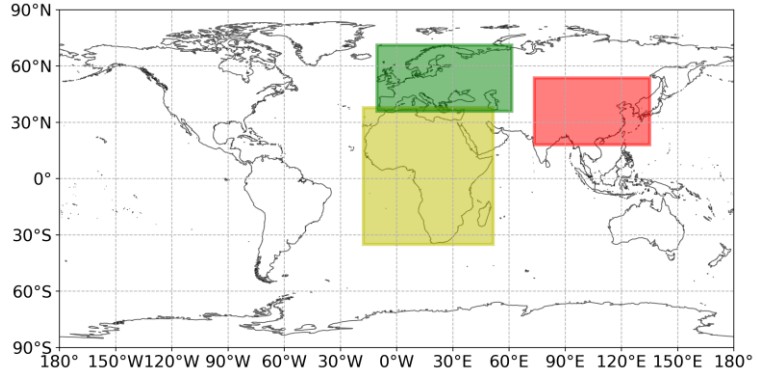

**Figure 1: Overview of the investigated regions Europe (green), China (red) and Africa (yellow) included in this study.**

In the first section (3.1), the differences between the cloud products based on TROPOMI version 1 and version 2 data are presented, focusing on the changes due to the version updates. Section 3.2 presents the results of the intercomparison between the cloud fractions and the cloud heights of the different cloud products using TROPOMI version 2 data. First, the cloud fractions are compared in respect of their correlations (3.2.1) and then are further analysed with respect to snow- and ice-

covered scenes over the region of Europe (3.2.2) and for scenes with sun glint in the Africa region (3.2.3). In Section 3.2.4, the results of the comparisons between the cloud heights are discussed, and across-track dependencies for both cloud fractions and cloud heights are shown in Section 3.2.5.

### 3.1 Comparison between version 1 and version 2 cloud products

In July 2020, the operational TROPOMI cloud products were updated from version 1 to version 2. This paper discusses the

comparison between version 1 and version 2 cloud fractions from OCRA/ROCINN, FRESCO, the $NO_2$ fitting window (cf_fit), and VIIRS for the region of Europe and the spring day (4 April 2019), which represents a day with snow- and ice-covered scenes (Figure 2 and Figure 3). In addition, the results of the cloud height comparison for the Africa region and the summer day (30 June 2018) are presented (Figure 4). Additional scatter plots of the cloud fractions and the cloud heights for the other test days for the regions of Europe and Africa are shown in Appendix A and in Section S1 in the Supplement.

The ROCINN CAL and OCRA a priori cloud fractions have a similar distribution (Figure 2b and c), both being systematically larger in version 2 than in version 1. The differences between the two versions are much smaller for ROCINN CRB (Figure 2a), arguably because cloud albedo has also changed but in the opposite direction, and the cloud fraction is scaled with the cloud albedo here as described in Section 2.2. However, the version 2 cloud fractions have higher values than the version 1





cloud fractions for all three OCRA/ROCINN products, especially for the largest values, due to an instrument degradation

correction introduced in version 2. ROCINN CRB is distributed closer to the 1:1-line, which might result from a change in the

surface albedo in version 2. ROCINN CAL and OCRA a priori version 2 show many values of one, while version 1 is smaller,

which is probably mainly related to the adapted OCRA scaling in version 2, i.e., the cloud fractions slightly below one in

version 1 are more likely to be fully cloudy in version 2. The differences in values of zero for all three OCRA/ROCINN

products may be due to the change from the OMI-based OCRA clear-sky reflectance climatology in version 1 to the

TROPOMI-based clear-sky reflectance in version 2. As mentioned in Section 2.1.1, ROCINN is not triggered for cloud

fractions smaller than 0.05. This can be seen in Figure 2b, where ROCINN CAL version 1 exhibits a gap for cloud fractions

smaller than 0.05 because the OCRA a priori cloud fraction in ROCINN CAL is set to zero. In contrast, ROCINN CAL

version 2 shows cloud fractions smaller than 0.05 due to a change in the co-registration procedure of the satellite pixels in

version 2. The co-registration from BD3 (UV/VIS) to BD6 (NIR) for the OCRA a priori cloud fraction and vice versa from

BD6 to BD3 for the ROCINN parameters is present in both product versions. However, while version 1 used a simplified fixed

pixel shift, in version 2, an improved scheme using the TROPOMI static mapping tables provided by KNMI was implemented.

Therefore, in version 2, pixels containing, e.g., cloud edges, may have cloud fractions smaller than 0.05.

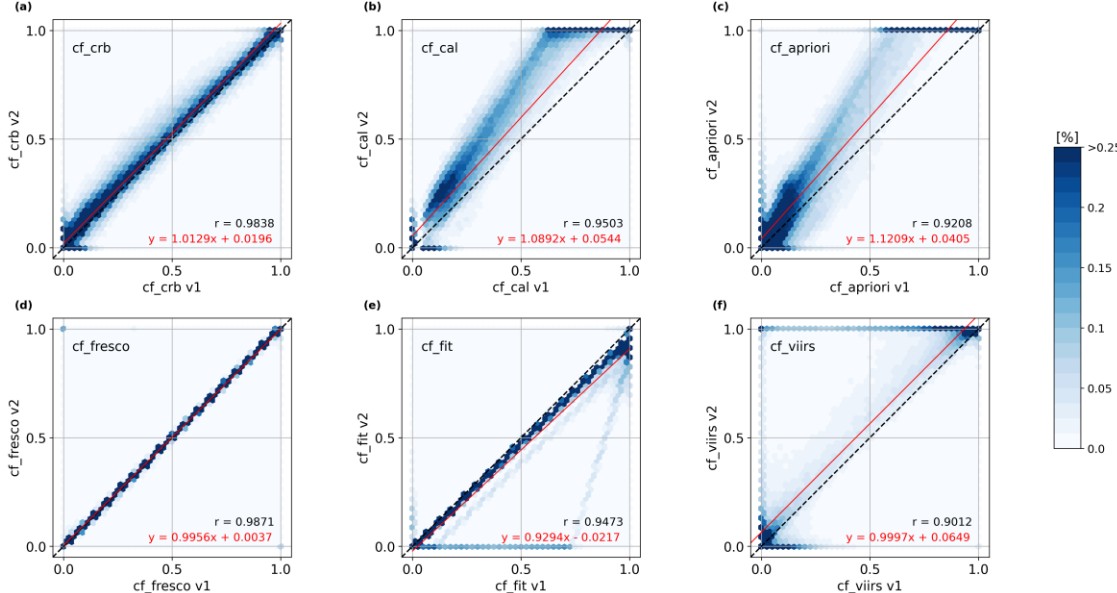

**Figure 2: Scatter plots between the version 1 and version 2 cloud fractions from (a) ROCINN CRB, (b) ROCINN CAL, (c) OCRA a**
**priori, (d) FRESCO, (e) the NO₂ fitting window, and (f) VIIRS for Europe and the spring day (4 April 2019).**

FRESCO shows virtually no change between the two versions for Europe (Figure 2d). However, some scatter is observed for

desert regions, such as Africa, for FRESCO cloud fractions smaller than 0.2 for the summer and winter days (Figure A1 and

Figure A2), likely resulting from a surface albedo adjustment in version 2 introduced to avoid negative cloud fractions. The

cloud fraction from the NO₂ fitting window changed for the largest values (Figure 2e), with version 2 being smaller than

version 1. This is due to adjustments in the cloud albedo to avoid cloud fractions larger than one and the use of degradation-





corrected irradiances in version 2, resulting in a higher irradiance signal and thus a lower reflectance. In addition, differences between the TROPOMI $NO_2$ product of version 1 and version 2 occur particularly over snow- and ice-covered scenes, where some additional lines below the 1:1-line in the scatter plot are found (Figure 2e), corresponding to positive differences larger than 0.3 on the map (Figure 3a). The reason for these differences is a change of the snow and ice mask from the Near-real-time Ice and Snow Extent (NISE) product in version 1 (Figure 3b) to a mask based on European Centre for Medium-Range Weather Forecasts (ECMWF) data in version 2. The ECMWF mask has a higher spatial and temporal resolution (Figure 3c). The VIIRS cloud fraction changed from VIIRS VCM (version 1) to ECM (version 2), resulting in large differences for the smallest and largest cloud fractions (Figure 2f).

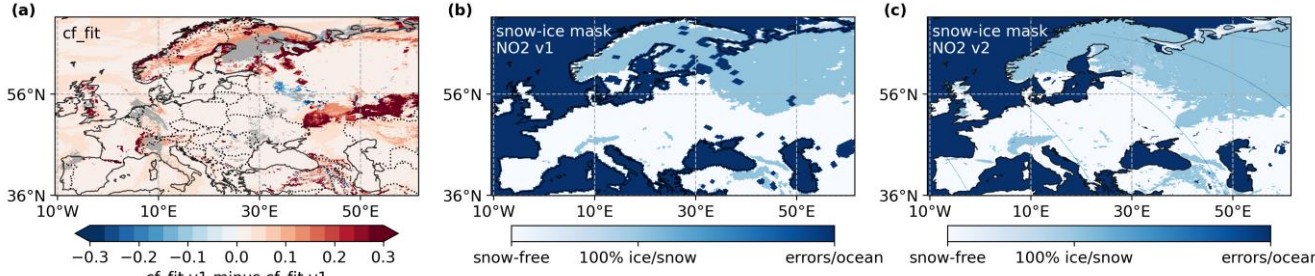

**Figure 3: Mapped differences between the cloud fractions from the TROPOMI $NO_2$ product of versions 1 and 2 for Europe and the spring day (4 April 2019) (a). Snow and ice mask from the TROPOMI $NO_2$ product of version 1 (b) and version 2 (c) for Europe and the spring day (4 April 2019). The version 1 map is from the NISE product, and the version 2 map is based on higher spatially and temporally resolved ECMWF data.**

Besides the cloud fractions, the cloud height from ROCINN CRB, the cloud top height from ROCINN CAL, and the FRESCO cloud height from the TROPOMI $NO_2$ product (ch_fresco*), which is remapped from the FRESCO NIR cloud pressure to the UV/VIS spatial footprints, were updated from version 1 to version 2. As the ROCINN CAL cloud base height is not a retrieved parameter but is only calculated with a constant geometric cloud thickness from the ROCINN CAL cloud top height, it is not included in this comparison.

In general, the ROCINN CRB cloud height (Figure 4a) and the ROCINN CAL cloud top height (Figure 4b) show smaller differences between the versions than the FRESCO cloud height (Figure 4c). This is also evident in the correlation coefficient; while ROCINN CRB and CAL show a correlation between the versions of 0.94 or 0.93, respectively, the FRESCO cloud heights correlate less well with a correlation of 0.75. Furthermore, the latter exhibit large scattering at heights lower than 2000 m and an additional vertical line of points for cloud heights where version 1 yields values close to 1000 m. One reason for the differences in the TROPOMI $NO_2$ products might be that in version 1, the cloud height converged to the surface pressure for low cloud fractions, while in version 2, this is less frequently the case. In addition, the error in the FRESCO cloud height becomes very large for small cloud fractions, and some of these pixels may still be within the reported error and contain small cloud fractions. However, these results seem to indicate that the cloud heights from the FRESCO cloud product converge in some cases with the ROCINN cloud heights in version 2, as the FRESCO cloud height in version 1 was found to be too low overall (Compernolle et al., 2021) and is partly larger in version 2, as shown by the vertical branch in Figure 4c.



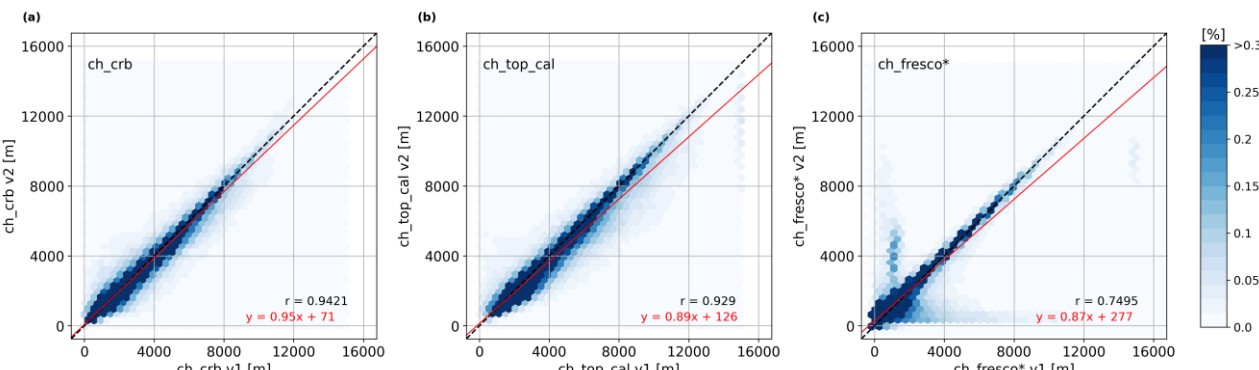

**Figure 4: Scatter plots between the version 1 and version 2 cloud heights from (a) ROCINN CRB, (b) ROCINN CAL top, and (c) FRESCO from the TROPOMI NO₂ product (ch_fresco\*) for Africa and the summer day (30 June 2018).**

## 3.2 Intercomparison between cloud products

As the different cloud products use different assumptions and algorithms, the results are not expected to always agree. For TROPOMI data users, it is therefore of value to understand the differences in the various cloud products and possible effects on trace gas retrievals.

The comparison of the different cloud fractions is always shown relative to the cloud fraction from the NO₂ fitting window (hereafter referred to as "cf_fit") and placed on the x-axis in the scatter plots. While this decision to use this cloud fraction as the reference is to some degree arbitrary, it is motivated by the fact that this is the cloud fraction currently used in the TROPOMI NO₂ product. As NO₂ is probably the most commonly used TROPOMI tropospheric trace gas product, it can be considered as the baseline. Consequently, the interpolated FRESCO cloud height from the TROPOMI NO₂ product (hereafter referred to as "ch_fresco\*") is used as a reference when comparing the different cloud heights to obtain consistent results. It should be understood that these are merely used as reference values and do not imply that they are the true values.

### 3.2.1 Cloud fraction

In this section, an intercomparison between the cloud fractions derived from the different version 2 cloud products is presented concerning statistics, such as correlations, to give an overview of their general behaviour. In the following sections, scatter plots and difference maps of the cloud fraction comparison are evaluated for specific situations, such as over snow and ice cover (3.2.2) and with sun glint (3.2.3) (see also Section S6 for more plots of the version 2 cloud fractions).

The tabular intercomparison of the correlations of the cloud fractions for the regions of Europe and Africa are shown in Figure 5 and Figure 6; see also Appendix B for the values of the slope and the y-intercept of the regression line of the scatter plots for Europe and Africa (see Section S2 for the statistics for China, and Section S3 for the version 1 data). As a first result, it can be said that the correlations between different products have improved in version 2 compared to version 1 (Section S3) for all regions and test days. The values for Europe (Figure 5) and China (Section S2) behave very similar, and therefore only the comparison for Europe is discussed in detail.





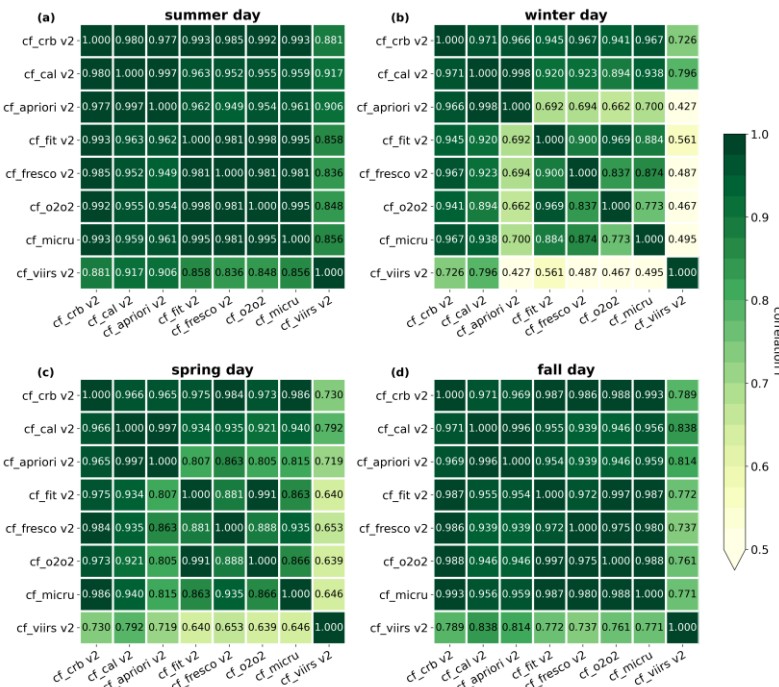


**Figure 5: Tabular intercomparison of the correlations between the version 2 cloud fractions from ROCINN CRB, ROCINN CAL, OCRA a priori, the NO₂ fitting window (cf_fit), FRESCO, O2-O2, MICRU, and VIIRS for Europe and (a) the summer day, (b) the winter day, (c) the spring day, (d) the fall day.**

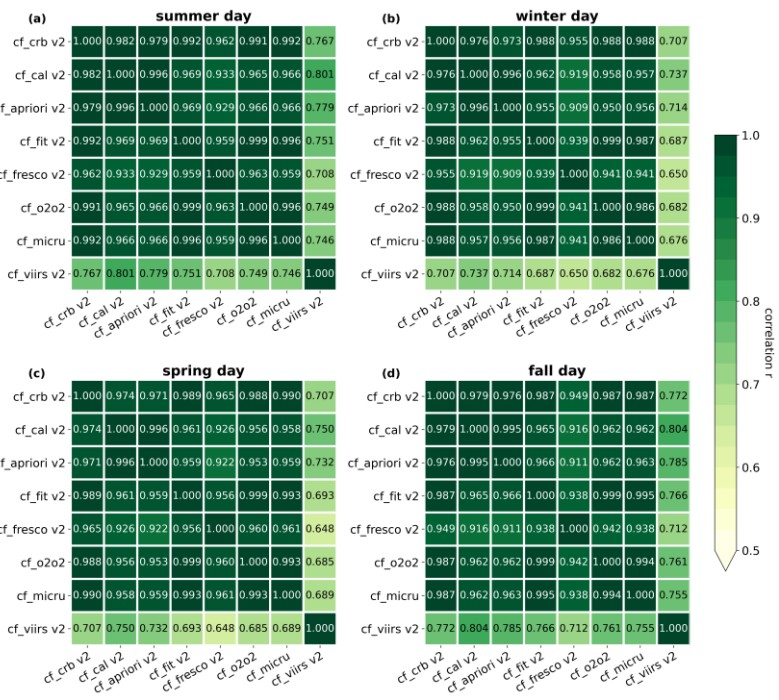

**Figure 6: As Figure 5 but for Africa.**





The summer and fall days for Europe exhibit overall better correlations, larger than 0.95, than the winter and spring days (Figure 5). This is especially true for the winter day, where the OCRA a priori cloud fraction deviates more from cf_fit, FRESCO, MICRU, and VIIRS than for other days, with correlations around 0.69. One reason for the poorer correlation between the cloud fractions for the winter and spring days is the product-specific treatment of snow and ice cover over Europe,

which is discussed in more detail in Section 3.2.2. The correlations with VIIRS are the worst, with the lowest values of 0.42 for the winter day and the largest values of 0.92 for the summer day, but they are still better than in version 1. It should be borne in mind that the VIIRS cloud fraction is a geometric cloud fraction, not an effective one like the cloud fractions from the other cloud retrieval algorithms; thus, these differences were expected.

For the region of Africa, high correlations of the cloud fractions of more than 0.9 are found for version 2 (Figure 6) as well as

version 1 (Section S3) for all days except for VIIRS, which is more different from the other products with correlations around 0.7 because VIIRS is a geometric cloud fraction. However, the variation of the values for the different days is minimal for Africa, which represents a desert region, in contrast to Europe, where the days show much more variations for the different seasons due to various influences such as snow and ice.

### 3.2.2 Cloud fraction over snow- and ice-covered scenes (Europe)

Snow and ice are a particular challenge for the cloud algorithms using UV-VIS-NIR (UVN) measurements, as the brightness of snow- and ice-covered surfaces is difficult to distinguish from optically thick clouds. As described in Section 3.1, the OCRA/ROCINN and FRESCO cloud products use snow and ice masks to determine the snow- and ice-covered scenes, which have a better spatial resolution in version 2 than in version 1 (see Figure 3b and c). Below, the different cloud fractions of version 2 are compared for the region of Europe and the spring day (4 April 2019), where snow and ice cover is present (Figure

7 and Figure 8). First, the general behaviour of the different cloud fractions compared to cf_fit is described for the Europe region (see Section S6 for the other test days), and second, the differences between the cloud products that occur due to snow and ice cover are discussed.

In general, the ROCINN CRB cloud fraction is larger than cf_fit, with a difference of about 0.1 for the largest values (Figure 7a). ROCINN CAL and OCRA a priori are clearly different from ROCINN CRB due to the scaling of the latter with the cloud

albedo (see Section 2.2). They show larger cloud fractions than cf_fit by up to 50% for the largest values, and little scatter for cloud fractions smaller than 0.2 (Figure 7b and c). FRESCO finds mainly larger values than cf_fit with a constant offset of about 0.1, and the points for smaller values are distributed in a hook shape (Figure 7d). O2-O2 and MICRU fit cf_fit very well (Figure 7e and f), only at the largest values where cf_fit is up to 30% smaller, and for MICRU, values lower than 0.1 scatter a little more. The O2-O2 cloud fraction and cf_fit are expected to have a relatively good agreement because the approaches are very similar. There are two subtle differences: First, for version 2, cf_fit is corrected for trace gas absorption in the $NO_2$ fitting

window, i.e., cf_fit is derived from the expected reflectivity without trace gas absorption using the $NO_2$ fit information (polynomial and Ring term) instead of the full reflectivity. This leads to a bit higher values for cf_fit than for O2-O2 since the trace gases absorb some of the light. Secondly, the wavelengths are quite close, but the albedo map evaluated for different





wavelengths is used to reflect the wavelength difference between the NO$_2$ and the O2-O2 fitting window. VIIRS shows mostly

larger values than cf_fit and has many values of 1 (fully cloudy) when cf_fit is smaller (Figure 7g), resulting from the strict

definition of cloudy pixels and the fact that it is a geometric cloud fraction. Larger scatter at the lowest 10-20% of the cloud

fraction values is found for all products except O2-O2. As a cloud fraction threshold of 0.2 is often used for trace gas retrievals,

such as NO$_2$, this result is particularly relevant to the application of cloud products to trace gas retrievals.

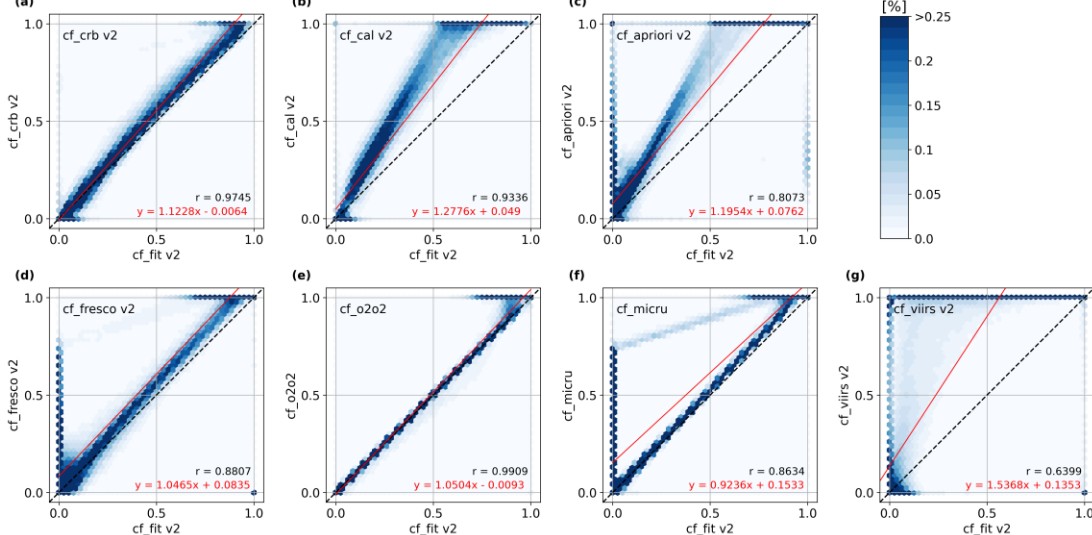

**Figure 7: Scatter plots between the version 2 cloud fractions from (a) ROCINN CRB, (b) ROCINN CAL, (c) OCRA a priori, (d) FRESCO, (e) O2-O2, (f) MICRU, and (g) VIIRS and the cloud fraction from the NO$_2$ fitting window (cf_fit) for Europe and the spring day (4 April 2019).**

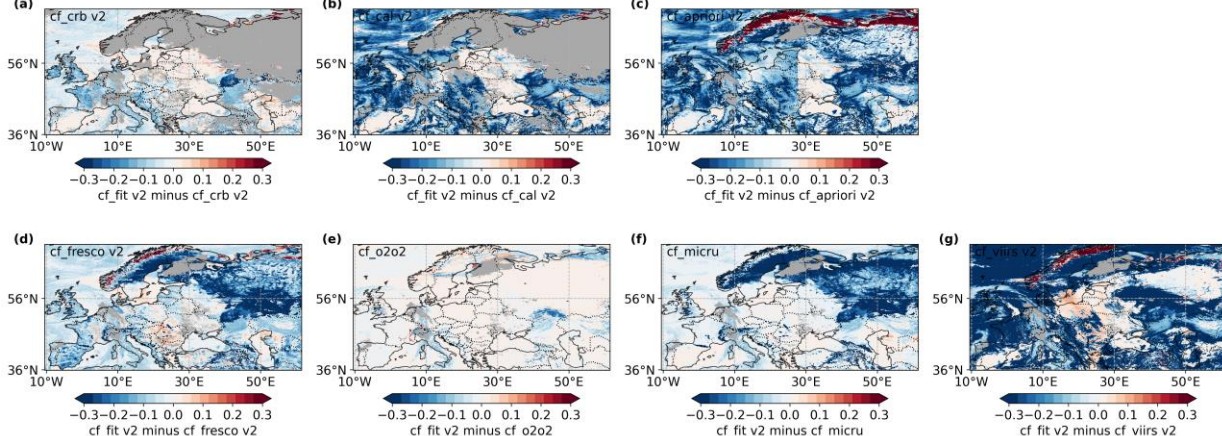

**Figure 8: Differences between the version 2 cloud fractions from (a) ROCINN CRB, (b) ROCINN CAL, (c) OCRA a priori, (d)**
**FRESCO, (e) O2-O2, (f) MICRU, and (g) VIIRS and the cloud fraction from the NO$_2$ fitting window (cf_fit) for Europe and the spring day (4 April 2019).**

Looking at the effect of snow and ice on the cloud products, specific differences between the cloud fractions are found. OCRA

a priori, FRESCO, MICRU, and VIIRS compared to cf_fit show an accumulation of extreme values (Figure 7c, d, f, g). These





values of the clusters correspond to negative differences larger than -0.3 over snow- and ice-covered regions such as Scandinavia and western Russia (Figure 8c, d, f, g). Some positive values larger than +0.3 are found over Norway for OCRA a priori, FRESCO, and VIIRS, but this is the exception. These differences occur because, unlike the other products, cf_fit detects clouds over Norway even though the cloudy pixels of the TROPOMI $NO_2$ product overlap areas identified as covered by snow and ice in the snow and ice mask of the TROPOMI $NO_2$ product (Figure 3c). This shows that the product's different snow and ice cover treatment can lead to large differences in the cloud fractions.

Contrary to OCRA a priori, FRESCO, MICRU, and VIIRS compared to cf_fit, CRB and CAL do not show an accumulation of extreme values in the scatter plots (Figure 7a and b). The reason is that in ROCINN CRB and CAL snow- and ice-covered pixels are filtered out mainly based on retrieval diagnostics, which significantly reduce the quality value for such challenging retrievals to at least 0.25. Consequently, pixels with snow and ice cover in CAL and CRB are not included in the comparison since only values with a quality value larger than or equal to 0.5 are used. These two products flag only 54% (CRB) and 57%
(CAL) of all values as valid in the region of Europe, as shown by a large number of grey pixels (NaN values) on the maps (Figure 8a and b). In contrast, the other cloud products include about 93% of the values because they do not flag snow and ice as strongly as ROCINN does (see Appendix D for all numbers of available values). For example, the MICRU algorithm treats snow- and ice-covered pixels the same as pixels free of snow and ice, resulting in overestimated cloud fractions over areas with variable snow and ice cover. As a result, MICRU compared to cf_fit exhibits a second correlation line for cloud fraction
values larger than 0.8 (Figure 7f). O2-O2 seems to be the only algorithm that treats snow and ice exactly like the TROPOMI $NO_2$ product, as it mostly shows excellent agreement with the cf_fit values, especially for smaller values (Figure 7e). The large differences between VIIRS and cf_fit are not mainly due to snow and ice, but the many VIIRS values of 1 (fully cloudy) resulting from the strict definition of cloudy pixels and the fact that the VIIRS cloud fraction is a geometric cloud fraction rather than an effective one like the cloud fractions of the other cloud products (Figure 7g and Figure 8g).

### 3.2.3 Cloud fraction with sun glint (Africa region)

The results of the test days for the region of Africa do not differ considerably, as mentioned in Section 3.2.1, and the distributions of the scatter plots between the different cloud fractions compared to cf_fit are virtually the same for all four days. The scatter plots and difference maps for the winter day (5 January 2019) are shown in Figure 9 and Figure 10, respectively, to present the differences that generally occur between the cloud products for the Africa region and to provide a
closer look at sun glint effects (see also Section S6 for the other test days).

The distributions of ROCINN CRB, ROCINN CAL, OCRA a priori, and VIIRS for Africa exhibit overall larger values than cf_fit (Figure 9a, b, c, g), as already seen for Europe (Figure 7). For FRESCO, the scattering points have a hook-shaped distribution, and the FRESCO cloud fractions are up to 0.5 larger for values where cf_fit is zero (Figure 9d). It should be mentioned that the surface albedo used by FRESCO in the NIR is not very appropriate for TROPOMI since it is derived from
GOME-2, and especially over vegetation, there are large systematic uncertainties with strong viewing angle dependence. The O2-O2 cloud fraction and cf_fit agree well for smaller values but diverge slightly for larger values where cf_fit is smaller





(Figure 9e). The latter is also true for the MICRU cloud fraction and cf_fit, and in addition, there is some scatter at cloud fraction values lower than 0.2 (Figure 9f), which is due to the different treatment of sun glint over water surfaces in the cloud retrieval algorithms.

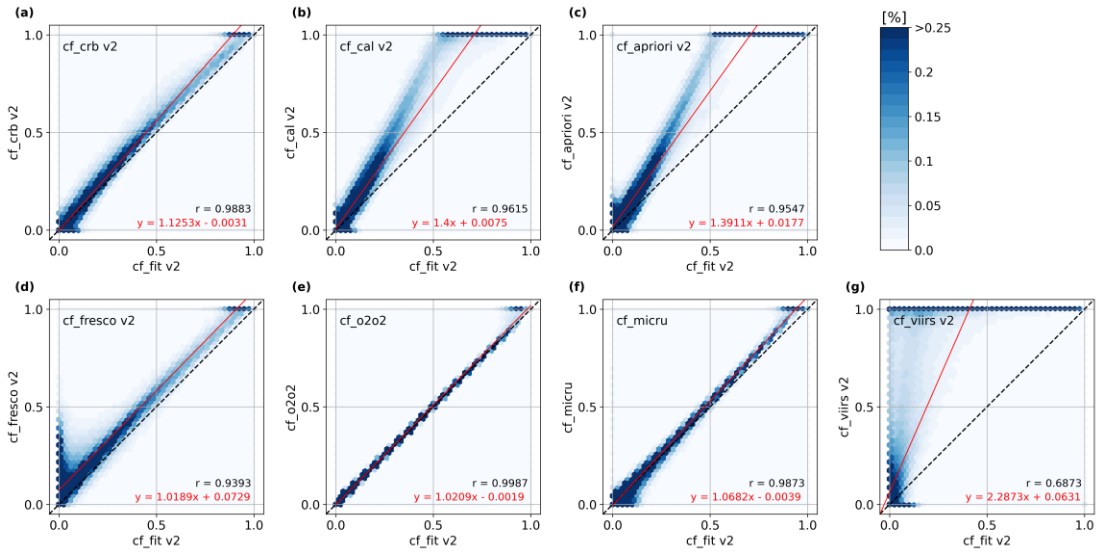


**Figure 9: Scatter plots between the version 2 cloud fractions from (a) ROCINN CRB, (b) ROCINN CAL, (c) OCRA a priori, (d) FRESCO, (e) O2-O2, (f) MICRU, and (g) VIIRS and the cloud fraction from the NO₂ fitting window (cf_fit) for Africa and the winter day (5 January 2019).**

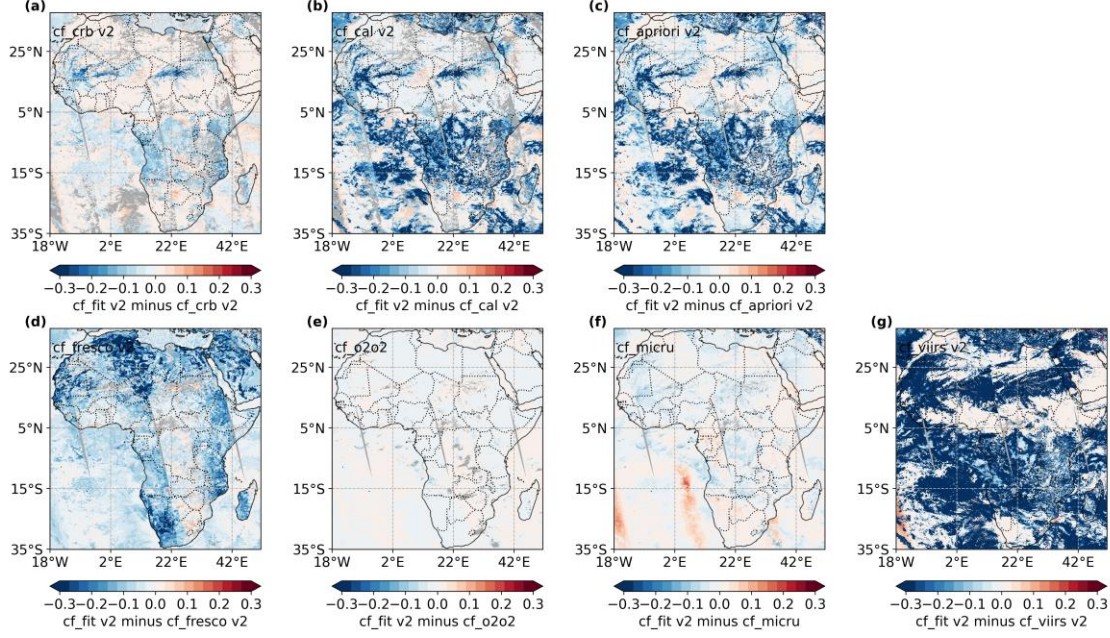


**Figure 10: Differences between the version 2 cloud fractions from (a) ROCINN CRB, (b) ROCINN CAL, (c) OCRA a priori, (d) FRESCO, (e) O2-O2, (f) MICRU, and (g) VIIRS and the cloud fraction from the NO₂ fitting window (cf_fit) for Africa and the winter day (5 January 2019).**



Sun glint affects satellite measurements when sunlight is reflected directly from the ocean surface to the sensor. In such cases, the otherwise dark ocean water is perceived as a bright surface, which can be misinterpreted as clouds due to high reflectivity

signals. The magnitude of the effect as well as the area affected depends on the smoothness of the surface and thus on wind speed. MICRU explicitly treats sun glint, which leads to differences from all the other products, as they do not have a specific treatment for sun glint. This can be seen in the fact that only the difference map of MICRU and cf_fit exhibits stripes over the oceans, here part of the Atlantic Ocean, where the sun glint geometry is given (Figure 10f). This corresponds with the cloud fraction map from the TROPOMI $NO_2$ product detecting apparent cloud veils over these areas, and with the scattering at the

lowest 0.2 cloud fraction in the scatter plots (Figure 9f). The different treatment of sun glint effects is also found on the difference maps for Africa for the other test days (Section S6). In this regard, it can be concluded that the cloud fractions over water surfaces might be most accurate in the MICRU algorithm.

### 3.2.4 Cloud height

The cloud height (CH), together with the cloud fraction, is an important parameter when comparing the different TROPOMI

cloud products to investigate the cloud impact on trace gas retrievals. In the following, the results of the comparison between the remapped FRESCO CH from the TROPOMI $NO_2$ product (ch_fresco*) and the ROCINN CAL cloud top height (CTH), ROCINN CAL cloud base height (CBH), the ROCINN CRB CH, and the O2-O2 CH for the region of China are presented. It should be noted that in the ROCINN CAL model, only the CTH is a retrieved parameter, while the CBH is assumed to have a fixed offset of 1 km from the CTH. ROCINN CAL CTH is expected to be higher than ch_fresco* because they are closer to

the geometric cloud edges than the cloud centroid height from the FRESCO algorithm. ROCINN CRB CH should agree well with ch_fresco* as the two algorithms are very similar in their approach. O2-O2 is the only algorithm that uses $O_4$-absorption in the VIS spectrum; thus, the height is expected to be the most different from the other products.

The tabular intercomparisons of all cloud products regarding the correlations are shown in Figure 11; the values of the slopes and the y-intercepts of the scatter plots for China can be found in Appendix C (see Section S4 for the statistics of Europe and

Africa, and Section S5 for the version 1 data). A good agreement of correlations is found for the summer and fall days between the OCRA/ROCINN products and ch_fresco* with values of about 0.8. However, for the winter and spring days, the correlations between CRB CH and ch_fresco* show smaller values of about 0.7 due to snow and ice cover in the region. The correlations of version 2 are mostly better than the correlations of version 1 (Section S5). The O2-O2 CH does not correlate well with the cloud heights from the other cloud products, with the worst correlations of around 0.5 for the winter and spring

days, as expected due to the different approach.





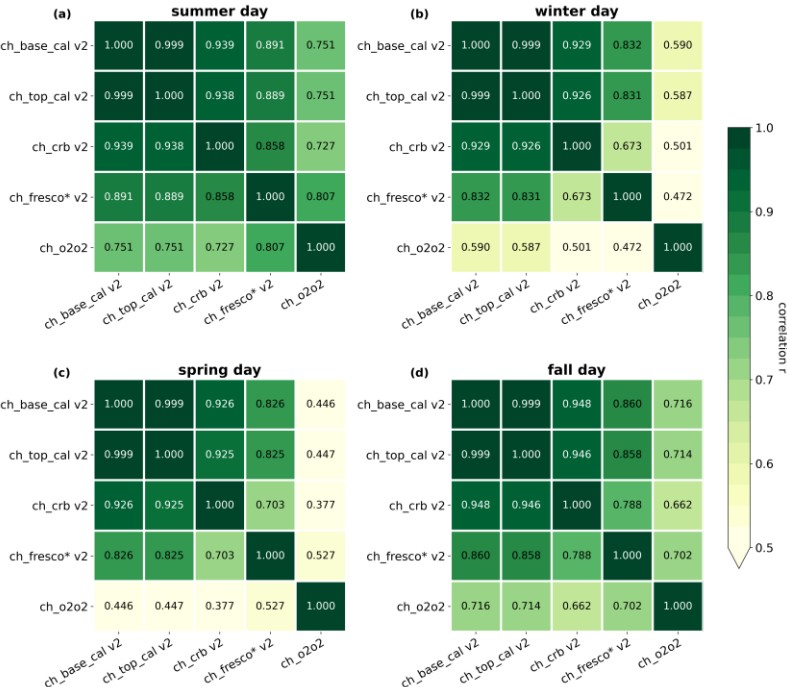

**Figure 11: Tabular intercomparison of the correlations between the version 2 cloud heights from ROCINN CAL base and top, ROCINN CRB, FRESCO from the TROPOMI NO$_2$ product (ch_fresco*), and O2-O2 for China and (a) the summer day, (b) the winter day, (c) the spring day, (d) the fall day.**

The cloud heights are evaluated explicitly for China and the fall day (20 September 2019) to provide an overview of the differences between the cloud products (Figure 12 and Figure 13; see Section S7 for scatter plots of the other test days). The scatter plots of the ROCINN CAL CBH, CAL CTH, and CRB CH compared to ch_fresco* show overall less scatter in version 2 (Figure 12a, b, c), especially at lower values, than in version 1 (not shown). However, some scatter remains for values smaller than 8 km. Against the expectation, the distributions of ROCINN CRB CH and CAL CBH compared to ch_fresco* look very similar. ROCINN CRB CH and CAL CBH are generally lower than ch_fresco* for higher clouds but fit better for lower clouds, while ROCINN CAL CTH is, on average, slightly larger than ch_fresco* for the lowest clouds and fits well for the highest clouds, as expected. This is also reflected in predominantly positive and negative differences, respectively, on the difference maps (Figure 13a, b, c) with deviations up to ±5 km, while in version 1, the deviations were up to ±8 km (not shown). The O2-O2 CH and ch_fresco* show larger scatter above and below the 1:1-line at cloud heights lower than 8 km (Figure 12d). All days show a stripe at the largest values of O2-O2 CH around 16 km when ch_fresco* has lower values, e.g., between 4 km and 8 km. The distributions of the differences between O2-O2 CH and ch_fresco* on the map have their own characteristics when all days are considered, and no regularity in their occurrence is found (see Figure 13d).

Finally, it should be noted that the ROCINN products CAL CBH and CTH, as well as CRB CH, compared to ch_fresco* have much fewer available values, only 43% and 44% of all values for the fall day and the region of China, respectively, than the FRESCO and O2-O2 products with 94% and 96%, respectively (see Appendix D for more details on the number of available


values). This limitation of values in ROCINN is not only due to snow- and ice-flagging, because only a small area is covered with snow and ice on that day, but the cloud heights are only available for pixels with a cloud fraction value above a threshold of 0.05. This generally leads to a small number of available cloud height values for ROCINN CAL and CRB.

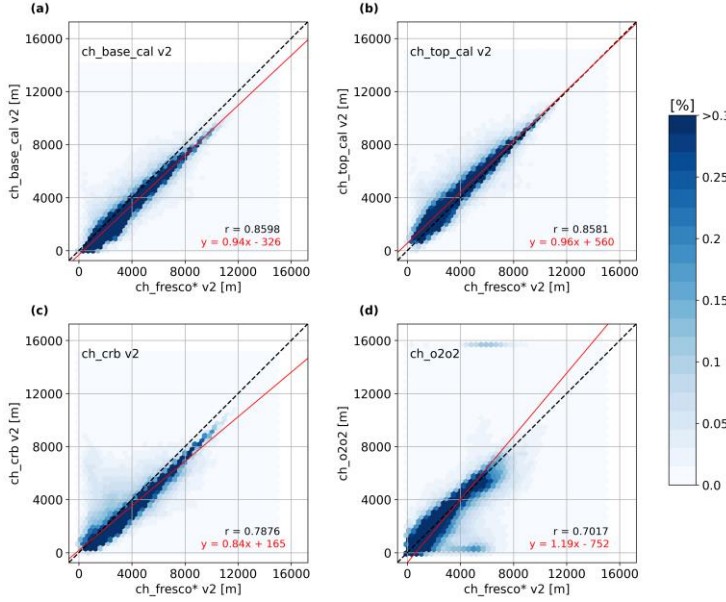

**Figure 12: Scatter plots between the version 2 (a) ROCINN CAL CBH, (b) ROCINN CAL CTH, (c) ROCINN CRB CH, and (d) O2-O2 CH and the FRESCO CH from the TROPOMI NO₂ product (ch_fresco*) for China and the fall day (20 September 2019).**

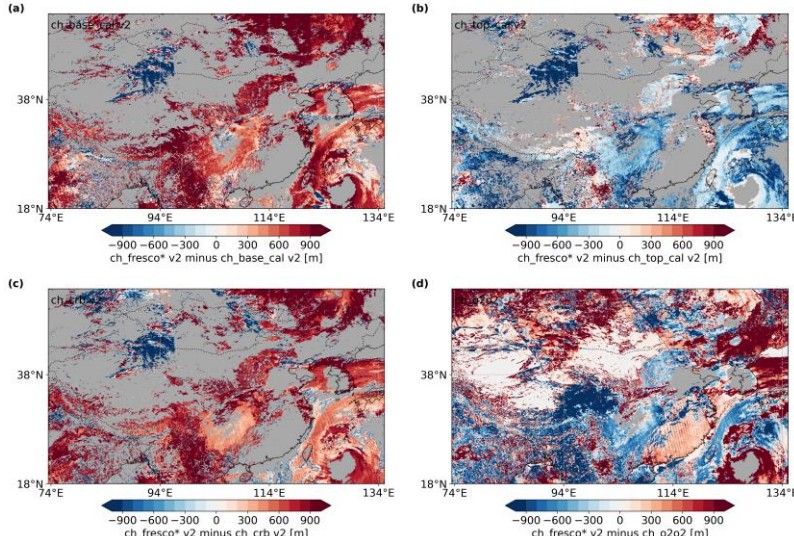

**Figure 13: Differences between the version 2 (a) ROCINN CAL CBH, (b) ROCINN CAL CTH, (c) ROCINN CRB CH, and (d) O2-O2 CH and the FRESCO CH from the TROPOMI NO₂ product (ch_fresco*) for China and the fall day (20 September 2019).**

As mentioned in Section 3.2.2, a cloud fraction threshold of 0.2 is often used for tropospheric trace gas retrievals because for larger cloud fractions, the information content on the lower troposphere is small. Therefore, this cloud fraction criterion is





applied in Figure 14 which only includes those scenes from Figure 12 having a cloud fraction less than or equal to 0.2 (see Section S7 for scatter plots of the other test days). Much more scattering occurs for the ROCINN products CAL CBH and CTH, as well as CRB CH compared to ch_fresco*, especially for cloud heights lower than 8 km (Figure 14a, b, c). This is

expected, as less information on cloud height is available at low cloud fractions. The distribution of the O2-O2 CH and ch_fresco* scatters mostly at cloud heights smaller than 4 km (Figure 14d). However, the stripes at the highest (about 16 km) and the lowest O2-O2 CH values seen for cloud heights without a limitation (Figure 13d) remain.

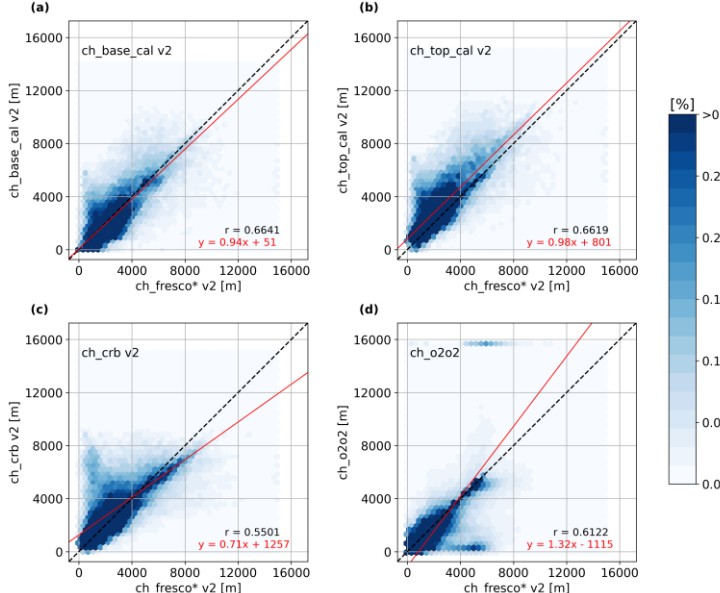

**Figure 14: As Figure 12 but with a cloud fraction threshold of 0.2 (only CH for scenes with cloud fractions ≤ 0.2).**

In addition to limiting cloud fractions to a threshold of 0.2, low cloud heights smaller than 2 km are particularly critical for tropospheric trace gas retrievals. In Figure 15, these two restrictions are applied to the cloud heights for the fall day and the China region, and some patterns can be seen (see Section S7 for the other test days). First, the ROCINN products compared to ch_fresco* scatter extremely at this lowest range of cloud heights with some clusters at different heights, e.g., from 0 m to 800 m for ROCINN CAL CBH and from 400 m to 1 km for ROCINN CRB CH, for which the regression line fits the perfect

line well. While the clusters for these two cloud products are below the 1:1-line, the ROCINN CAL CTH shows even two clusters at different heights, one above and one below the 1:1-line from about 600 m to 1800 m. The O2-O2 CH and ch_fresco* largely agree for values larger than 1 km. However, for values lower than 1 km, ch_fresco* is mainly larger than the O2-O2 CH, and a second branch and much scattering occur.

These results show that the cloud heights of the different cloud retrieval algorithms differ significantly when only scenes with

low cloud fraction are considered. When only the lowest cloud height values are considered, which are critical for tropospheric trace gas retrievals, the scatter further increases. Differences between the cloud heights without these limitations appear to be acceptable.





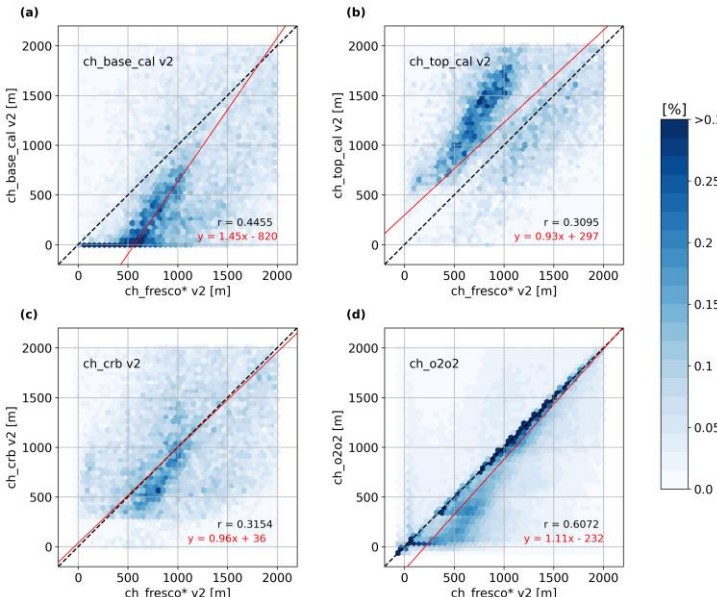

**Figure 15: As Figure 14 but only for cloud heights lower than or equal 2 km (only CH ≤ 2 km for scenes with cloud fractions ≤ 0.2).**

### 3.2.5 Across-track dependencies

For many products of nadir viewing UV/VIS instruments, across-track biases have been reported. Possible reasons are instrumental effects, radiative transfer effects such as surface BRDF or the angular dependency of the scattering phase functions, or observational effects, e.g., from the three-dimensional structure of clouds. No observational effects would be

expected for full cloud cover and cloud-free pixels for both cloud fraction and cloud height. However, for partially cloudy pixels, a systematic effect with higher apparent cloud fractions would be expected for large observation angles, which can be explained by the cloud holes appearing smaller for slant light paths. Moreover, for partially cloudy pixels, a small systematic effect with possibly lower apparent cloud heights could be expected, which may result from the fact that the sides of the clouds contribute more to the measured signal for slant viewing angles. In addition to systematic effects, relatively small data samples

may also have across-track variations from the specific sampling of the scene used.

In the following, across-track dependencies of the different version 2 cloud products are shown in line diagrams in which the daily averaged cloud fractions and cloud heights are plotted against the across-track index of the orbits (see Section S8 for the plots of version 1). The values are flagged for snow and ice to ensure a consistent analysis for the different days. In addition, only pixels for which all products have valid values are included in daily means because the OCRA/ROCINN products

compared to the TROPOMI NO$_2$ product have significantly fewer valid values than the other products compared to the TROPOMI NO$_2$ product (see Appendix D for detailed tables). This is especially true for the 22$^{nd}$ across-track index, where a dip occurs in all cloud fraction and cloud height diagrams (e.g., Figure 16 to Figure 21); its cause is discussed further below.





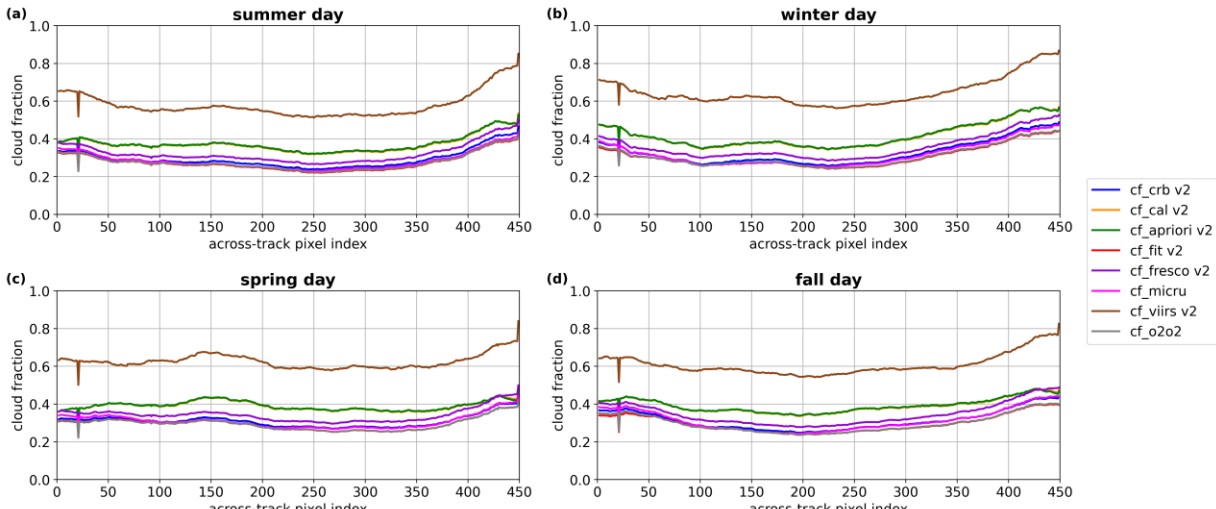

**Figure 16: Mean values of version 2 cloud fractions, as a function of the across-track index, from ROCINN CRB, ROCINN CAL, OCRA a priori, the NO₂ fitting window (cf_fit), FRESCO, MICRU, VIIRS, and O2-O2 for the globe and (a) the summer day, (b) the winter day, (c) the spring day, and (d) the fall day with quality- and snow-/ice-flagging, and including only pixels having valid values for all products.**

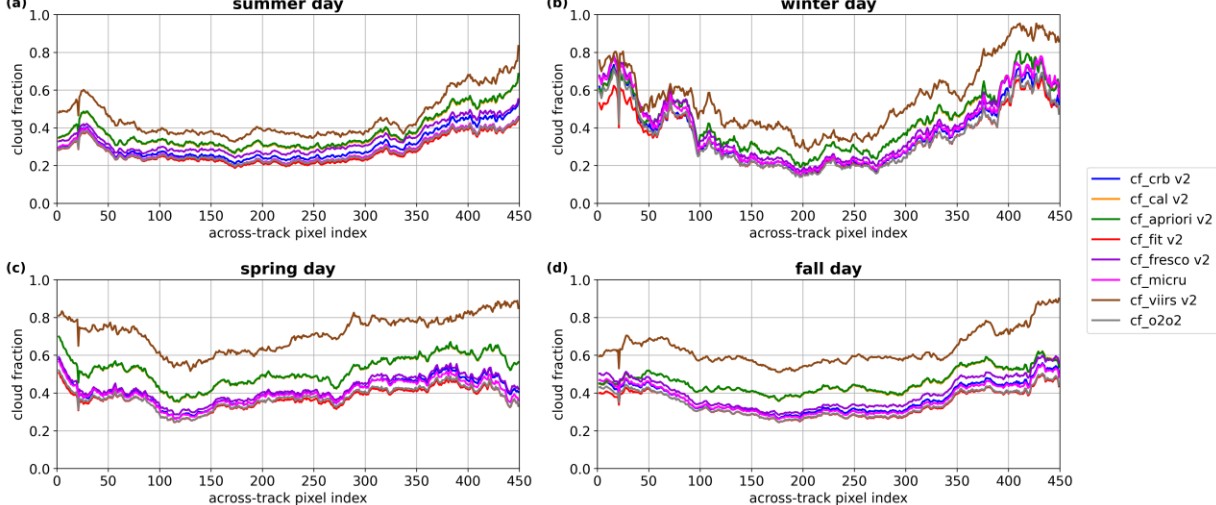

**Figure 17: As Figure 16 but for Europe.**

For the regions of Europe and China, the cloud fraction plots differ when comparing the four days, and the number of included values varies largely with about 77% and 81% of all pixels for the summer day, 20% and 45% for the winter day, 50% and 65% for the spring day, and 80% and 77% for the fall day, respectively (see Appendix D). For Africa, the number of valid values is more constant, with about 81-88% of all pixels for the different days. The same is true for the globe as a whole, where about 60% of the pixels are included in daily mean cloud fractions for the four days. The line plots for the globe (Figure 16) and for Africa (Figure E1) are very similar for the different days. VIIRS generally shows the largest values compared to the





other products (up to 20-40% larger). However, all products exhibit slightly larger values for the largest across-track pixel indices, and the curves look U-shaped, as one would expect from geometrical considerations for broken cloud fields. The OCRA a priori and ROCINN CAL are virtually identical. For Africa (Figure E1), FRESCO is smaller than these two products on the left-hand side of the indices and larger on the right-hand side, which points to a BRDF effect in the FRESCO surface

reflectivity. The cf_fit, O2-O2, MICRU, and ROCINN CRB have a very similar run of the curves. Globally (Figure 16), FRESCO is always smaller than OCRA a priori and ROCINN CAL, but larger than the other products, which overlap more in the centre of the orbit and diverge slightly at the edges. For Europe (Figure 17) and China (Figure E2), the line plots for the winter day are strikingly different from the other days, showing larger cloud fractions at the extreme values of the across-track pixel indices and up to 60% smaller values at the centre.

It should be noted that the overall good consistency of the plots is only obtained by filtering the values (snow and ice flag and overlap of available values in all products). When using all valid values for each product independently, quite different curves are obtained for those products having many more valid values. As an example, the global diagrams are shown in Figure 18. Without the filtering, the cloud fractions of ROCINN CAL and OCRA a priori are no longer identical, as already seen when comparing the cloud products with cf_fit in the scatter plots. Overall, the OCRA/ROCINN, FRESCO, cf_fit, O2-O2, and

MICRU curves have similar shapes but large offsets. In contrast, the offset between VIIRS and the other cloud products is not as large as when the data is filtered, with about 10-30%, and the VIIRS curves are less U-shaped. In addition, the non-filtered plots indicate that the above-mentioned dip in the curves at the 22nd across-track pixel index remains on the OCRA/ROCINN cloud products, while the other products show a smooth behaviour. This dip might be related to the changing binning scheme towards the swath edges, as the TROPOMI ground pixel size changes at this detector position. The weights from the co-

registration mapping tables also show this dip (not shown); hence it could also be a consequence of the co-registration treatment in the OCRA/ROCINN algorithm.

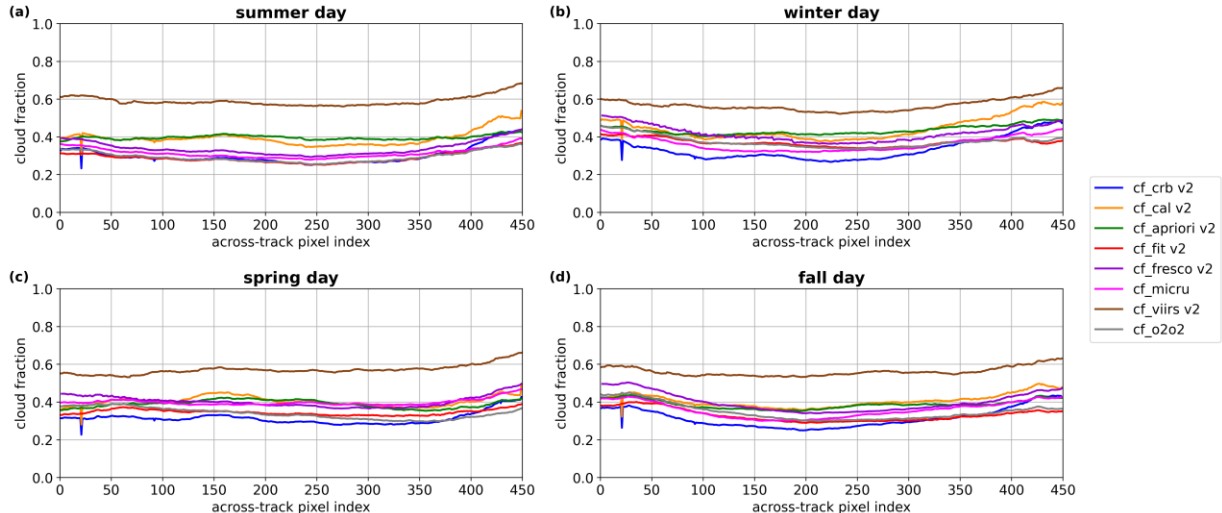

**Figure 18: As Figure 16 but data filtering (quality- and snow-/ice-flagging) is not applied, thus all valid values are included for each product independently.**





However, there is no overall indication for a systematic across-track problem in the cloud fractions of any of the products except for the above-mentioned FRESCO issue over Africa. The behaviour of the OCRA/ROCINN curves in version 2 is in general less different from that of the other products than in version 1 (Section S8). Consequently, it can be stated that the comparability of the cloud fractions has improved after the version update. However, it should be noted that the differences between the cloud products depend strongly on how they are compared.


The across-track dependencies for the cloud heights in global terms look very similar, with slightly bent curves and minima at the centre of the index range for all days (Figure 19, see Section S8 for version 1 plots). This result does not correspond to the three-dimensional geometrical consideration for clouds and the above-mentioned expectation that the cloud height might be systematically smaller for slant viewing angles, which would result in a maximum for the centre of the pixel indices a
maximum. However, the number of included values is very small, about 14-26% of all pixels. This is due to the fact that only pixels are used for which all retrievals result in a valid cloud height. The interpolated FRESCO CH from the TROPOMI $NO_2$ product (ch_fresco*) is mainly larger than the ROCINN CRB CH, the CAL CBH, and the O2-O2 CH but smaller than the ROCINN CAL CTH, with differences of about 500 m. O2-O2 differs more from ROCINN CRB for the spring and fall days, especially in ranges of lowest and largest across-track pixel indices. Both ch_fresco* and the O2-O2 CH show some steps in
the lines that are probably linked to interpolation in look-up tables. The distributions of the curves for Africa are comparable to those for the globe, but the curves are less smooth, and the difference between ch_fresco* and the other cloud heights, with the exception of the ROCINN CAL CTH, is overall smaller (Figure F1).

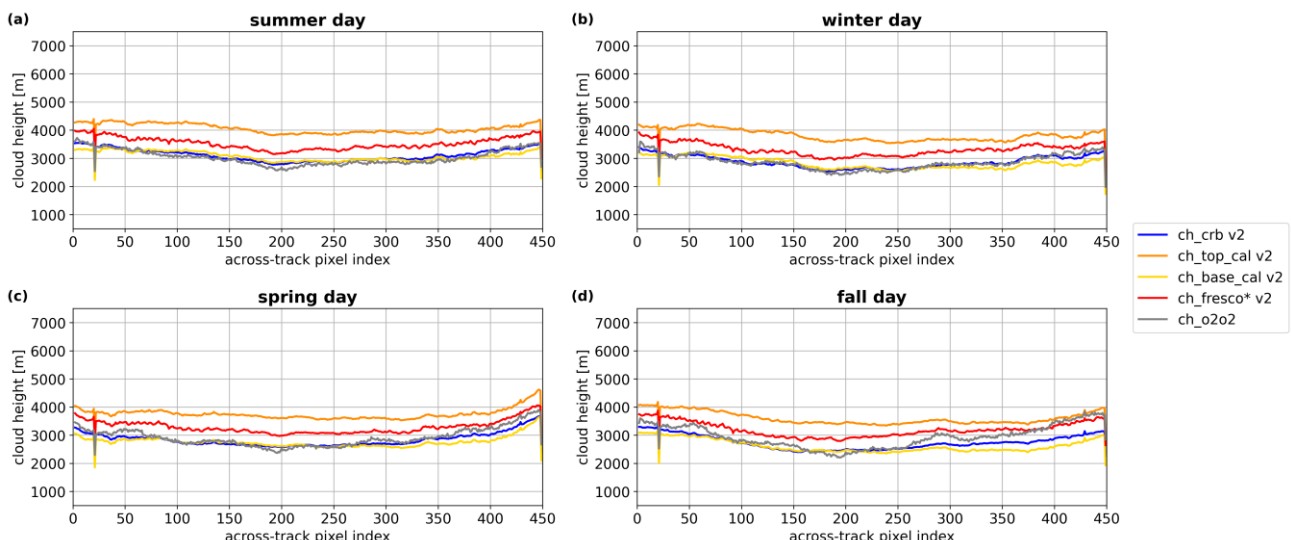

**Figure 19: Mean values of version 2 cloud heights, as a function of the across-track index, from ROCINN CRB, ROCINN CAL top
and base, FRESCO from the TROPOMI NO₂ product (ch_fresco*), and O2-O2 for the globe and (a) the summer day, (b) the winter
day, (c) the spring day, and (d) the fall day with quality- and snow-/ice-flagging, and including only pixels having valid values for all
products.**





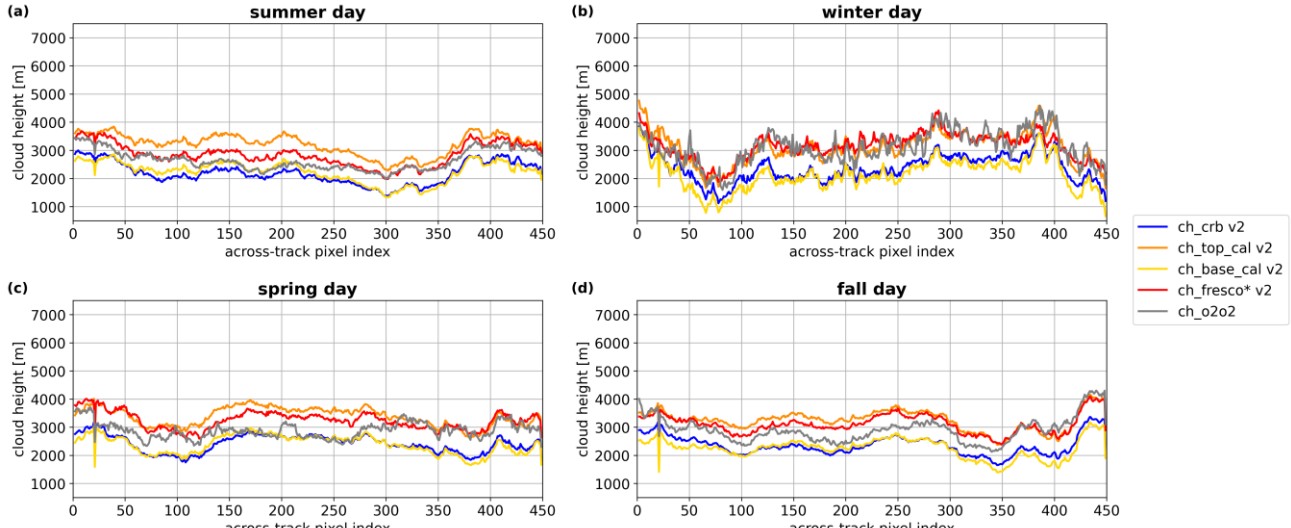

**Figure 20: As Figure 19 but for Europe.**

As noted for the cloud fractions, the variability of the across-track dependencies of the cloud heights for the different days is larger for Europe (Figure 20) and China (Figure F2) than for Africa and globally. For China and the summer day, ch_fresco* shows similar values as the ROCINN CAL CBH at the centre of the indices but larger values at the edges. ROCINN CRB and O2-O2 are smaller than these products. Overall, the ROCINN CAL CTH is the largest, but for the winter day, the products behave very strangely compared to the other days and regions with east-west differences in the pixel indices and bent curves that have their minima in the middle of the indices. O2-O2 has much larger values at the edges, and the ROCINN CAL CBH is overall the smallest height. The latter is also observed for the spring and fall days. For Europe, the winter day also shows east-west differences in the pixel indices, but in contrast to China, the curves exhibit the maxima on the left side of the pixel index and the minima on the right side. For all days, the ROCINN CRB CH and the CAL CBH are the smallest cloud heights, while ch_fresco* is in between the ROCINN CAL CTH and the O2-O2 CH.

It should be noted that the across-track mean diagrams can vary quite strongly from day to day, especially when only sub-regions are considered rather than a global distribution, and that across-track plots might look much smoother when weekly or monthly means of data are considered. In addition, as shown for the cloud fractions, the cloud products behave differently when the filtering conditions are changed. When all valid values for each product are used independently, the cloud height curves show a different arrangement than with filtering (Figure 21). While the ROCINN CAL CTH and CBH do not change

significantly, the ROCINN CRB CH is slightly larger than the ROCINN CAL CBH and lies partly in between the two ROCINN CAL cloud heights. O2-O2 shows no systematic features and seems to behave randomly, e.g., for the summer day, it is mostly smaller than ROCINN CAL CBH, for the winter, it is more like ROCINN CRB, for the spring day, it is between ROCINN CRB and CAL CTH, and for the fall day, it behaves more like ROCINN CAL CBH on the left side of the across-track indices and on the right side, it shows large values like ROCINN CAL CTH. Overall, the curves are skewed, the left side showing



lower values than the right side. The most unexpected changes are found for ch_fresco*, which exhibits values up to about 1 km lower than with filtering and therefore has values up to 500 m smaller than ROCIN CAL CBH, especially in the middle of the indices. This indicates that the finding that cloud heights from ROCINN and FRESCO are closer in version 2 only holds if the same filtering of the values is applied to all cloud products.

In summary, some systematic across-track problems in the cloud heights are evident, such as the unexpected minima of the

cloud heights in the centre of the pixel indices that cannot be explained by observational effects and the unexpected east-west differences of the cloud heights in the pixel indices for the winter day and the regions of China and Europe. In addition, O2-O2 shows larger variability than the other products and tends for larger values towards larger across-track indices in some seasons and scenarios. Overall, the agreement between the cloud products strongly depends on how the cloud fractions and cloud heights are filtered.

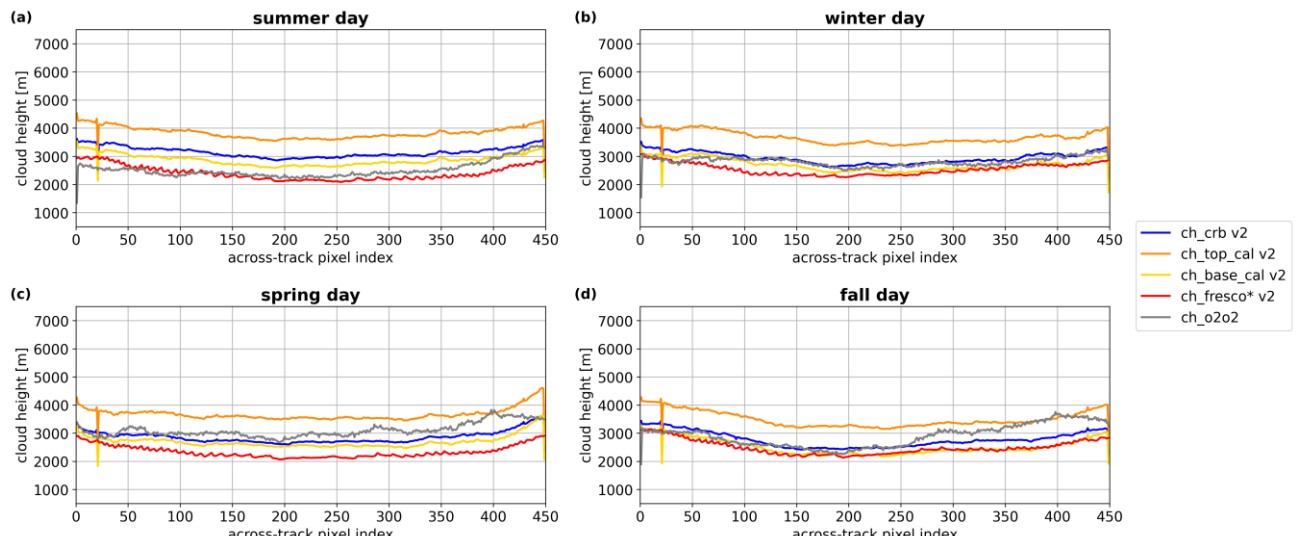


**Figure 21: As Figure 19 but data filtering (quality- and snow-/ice-flagging) is not applied, thus all valid values are included for each product independently.**

## 4 Conclusions

Cloud information is essential for quantitative retrievals of trace gas columns from UV/VIS satellite observations. This study

reports on a systematic comparison of different cloud products for the TROPOMI instrument on board the Sentinel-5 Precursor satellite. In a first step, the versions 1 and 2 of the TROPOMI cloud products ROCINN CRB, ROCINN CAL, OCRA a priori, FRESCO, the cloud fraction from the $NO_2$ fitting window, and VIIRS are compared. The cloud fractions from the OCRA/ROCINN cloud products show the largest differences between version 1 and version 2, the ROCINN CRB cloud fraction being less affected by the version change due to the scaling with the cloud albedo. The FRESCO product shows

virtually no changes, and the cloud fraction from the $NO_2$ fitting window changed principally over snow- and ice-covered scenes due to an update of the snow and ice mask. The VIIRS cloud fraction shows large differences at the smallest and the





largest values due to the version change from VIIRS VCM to VIIRS ECM. Concerning the cloud heights, the largest changes are found in the remapped FRESCO cloud height from the TROPOMI NO$_2$ product, while the ROCINN CRB and CAL cloud heights show merely small differences. Overall, the cloud heights from the different cloud retrieval algorithms converged, which is a good result as the FRESCO cloud height in version 1 was too low overall (Compernolle et al., 2021).

The second part of this work compares the above-named TROPOMI version 2 cloud products, as well as the O2-O2 product and the MICRU cloud fraction. Compared to version 1, general improvements include smaller scatter for small cloud fractions in version 2, and the fact that the across-track bias in the ROCINN cloud fractions found by Compernolle et al. (2021) in version 1 is no longer present in version 2. In addition, better comparability of the cloud products is found in version 2, resulting from improvements in the various retrievals. For OCRA/ROCINN, the following changes were applied: an instrument
degradation correction, a change in the surface albedo treatment (daily updated G3_LER retrieval from TROPOMI instead of fixed climatology), the adapted OCRA scaling, the change in the UV/VIS co-registration procedure in version 2, and the change from the OMI-based OCRA clear-sky reflectance climatology in version 1 to the TROPOMI-based clear-sky reflectance in version 2. While FRESCO for Europe effectively shows no changes between the two versions, the cloud fractions smaller than 0.2 for Africa differ slightly due to a surface albedo adjustment in version 2. Since the albedo is a major cause of
differences between the different cloud products, in the upcoming version 2.4.0 of the FRESCO product, a directional-dependent LER derived from TROPOMI observations will be used, which is expected to improve systematic biases in the FRESCO cloud fraction. In the TROPOMI NO$_2$ product of version 2, the cloud albedo is adjusted when the cloud fraction exceeds 1, and a degradation-corrected irradiance is used. In addition, the FRESCO cloud heights converge less frequently to the surface pressure for low cloud fractions in version 2 than in version 1. Furthermore, a spatially and temporally better
resolved snow and ice mask based on ECMWF data is implemented in all version 2 products, as opposed to the mask from the NISE product used in version 1.

The different TROPOMI cloud products of version 2 still show some systematic differences, both in the scatter plots of the cloud fractions and the comparisons of cloud heights. However, the variations between the different days are smaller than in
version 1. A large part of the differences can probably be explained by the different assumptions made by the cloud products regarding the cloud model used (e.g., Lambertian cloud or scattering cloud) and the surface albedo used (e.g., daily retrieved from TROPOMI or based on a fixed climatology). Another important source for differences is the behaviour under difficult surface conditions such as snow and ice cover and how the values for these situations are flagged in the cloud retrieval algorithms. Only one day per season is used in this study, but the same pattern of differences is seen when examining more
days; the results are not presented in this paper.

Summarizing the results for the region of Europe, the ROCINN CAL and the OCRA a priori cloud fractions are predominantly larger than cf_fit, while ROCINN CRB and FRESCO show a small offset of 5%, and the cloud fractions from the O2-O2, MICRU, and TROPOMI NO$_2$ products have the overall best agreement with small differences only for the largest cloud fractions. Additional systematic issues are found for snow- and ice-covered scenes, e.g., clusters of extreme values in the
OCRA a priori, FRESCO, MICRU, and VIIRS plots, as well as a second correlation line for MICRU and FRESCO. In





summary, the different treatment of snow and ice cover leads to large differences between the cloud products, as the cloud retrieval algorithms apply varyingly strict flagging of snow- and ice-covered pixels.

For Africa, the cloud fractions show essentially the same patterns as for Europe. Only the NIR-based FRESCO cloud fraction shows biases over vegetation due to a differently derived surface albedo. The MICRU cloud fraction shows more scatter at the

lowest values due to the explicit treatment of sun glint in the MICRU algorithm. Therefore, the MICRU cloud fraction is arguably more accurate over water surfaces affected by sun glint than the other cloud products that do not specifically treat sun glint in their algorithms.

To conclude the results of the cloud heights, overall better agreement is found between the ROCINN CAL and CRB products and the interpolated FRESCO cloud height from the TROPOMI NO$_2$ product (ch_fresco*) in version 2 than in version 1 with

correlations around 0.8. The ROCINN CAL cloud base height and the ROCINN CRB cloud height are systematically lower than ch_fresco*, in particular at large values. For cloud heights lower than 2 km and applying a cloud fraction threshold of 0.2 as often done in trace gas retrievals, the ROCINN products compared to ch_fresco* show overall large scatter and clusters of values at different heights. The number of valid values differs largely between the ROCINN products and the FRESCO CH, with the ROCINN products providing only half the number of values of the TROPOMI NO$_2$ product in some cases due to a

different flagging procedure of the cloud height values. The O2-O2 cloud height shows mainly smaller values than ch_fresco* and much scattering for low clouds, as well as two clusters of values at the largest and lowest cloud heights when ch_fresco* is between 4 km and 8 km. The larger differences between the O2-O2 cloud height and the other cloud products were expected because the O2-O2 cloud product is the only algorithm in this study that uses the O$_4$-absorption in the blue part of the spectrum. The across-track dependency plots of the different cloud products show slightly U-shaped curves with minima in the centre of

the across-track pixel indices for the cloud fractions and the cloud heights. As for the cloud fraction, these bent curves can be explained by geometrical considerations for partially cloudy pixels because higher apparent cloud fractions are expected for large viewing angles. For the cloud height, the expectation that the cloud height might be smaller for slant viewing angles due to the larger contribution of the cloud sides to the measured signal cannot be confirmed. In addition to these issues, the FRESCO cloud height from the TROPOMI NO$_2$ product and the O2-O2 cloud height show some steps in the curves, probably

related to the interpolation in the look-up tables. O2-O2 exhibits large variability and tends to have larger values at larger across-track indices in some seasons. In addition, the cloud heights show differences between the eastern and western pixels. Another systematic problem for the cloud fraction is that the FRESCO cloud fraction curves for the region of Africa are shifted due to a BRDF effect in the FRESCO surface reflectivity. Overall, the comparability of the cloud fractions has improved after the version update. However, the differences between the cloud products vary depending on how the cloud fractions and cloud

heights are compared. For example, the FRESCO cloud height from the TROPOMI NO$_2$ product is mainly larger than CRB, CAL cloud base height, and O2-O2 using the data filtering (snow/ice flag and data consistency), but it is the smallest cloud height without the data filtering. This should be kept in mind when considering the positive result of this study that the FRESCO and OCRA/ROCINN cloud heights are closer after the version update.



Taken as a whole, the different TROPOMI cloud products in version 2 are highly correlated. The differences between the
cloud products for Europe and China in terms of cloud fraction and cloud height are much larger than those for Africa when
comparing the different seasons due to the surface conditions such as snow and ice cover. Differences are larger at small cloud
fractions and for low clouds, situations relevant for tropospheric trace gas retrievals. When comparing version 2 and version 1
products, the consistency between the cloud products has significantly improved, which is an important message to TROPOMI
data users applying the cloud products for trace gas retrievals.






**Appendix A: Additional plots of the comparison between the version 1 and version 2 cloud fraction for Africa**

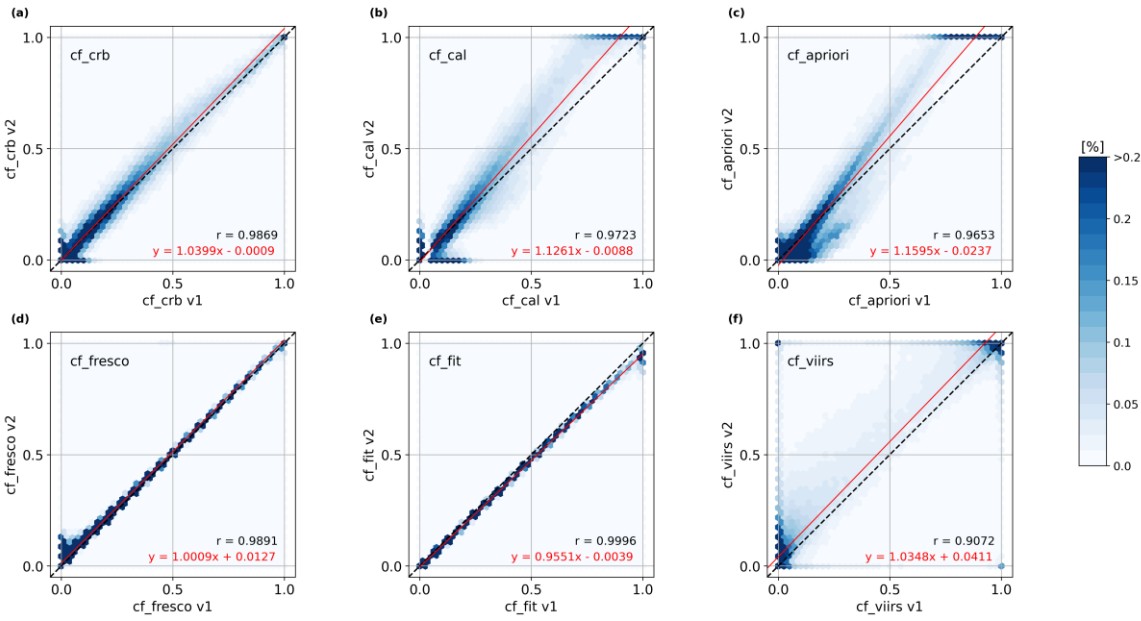

**Figure A1: Scatter plots between the version 1 and version 2 cloud fractions from (a) ROCINN CRB, (b) ROCINN CAL, (c) OCRA a priori, (d) FRESCO, (e) the NO₂ fitting window, and (f) VIIRS for Africa and the summer day (30 June 2018).**

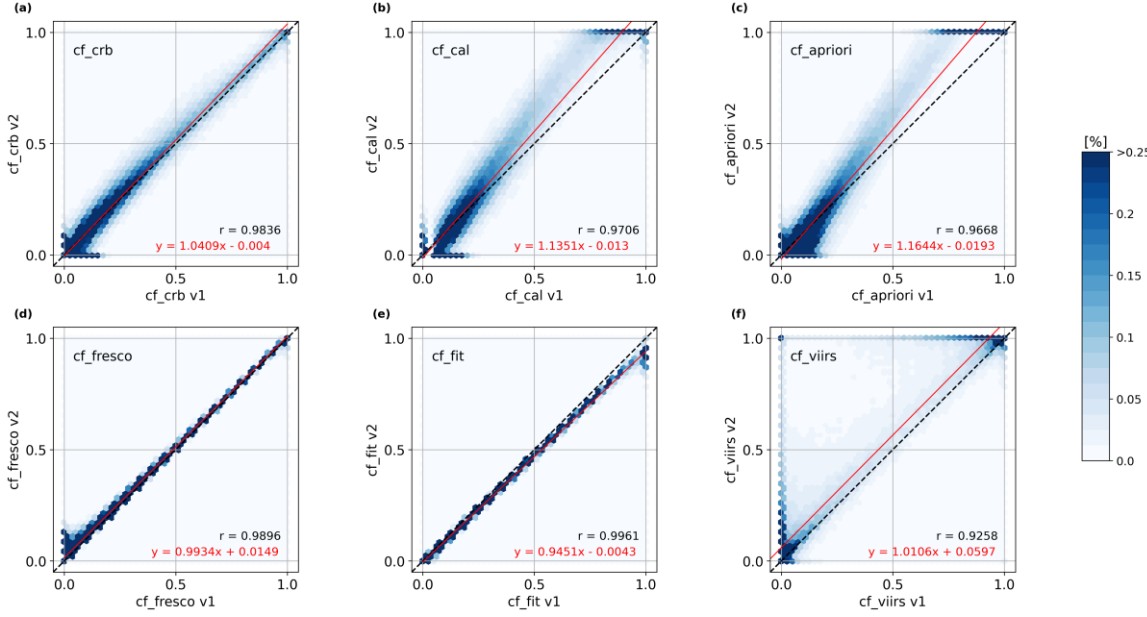


**Figure A2: As Figure A1 but for the winter day (5 January 2019).**





**Appendix B: Tabular intercomparison of the statistics between the version 2 cloud fractions for Europe and Africa**

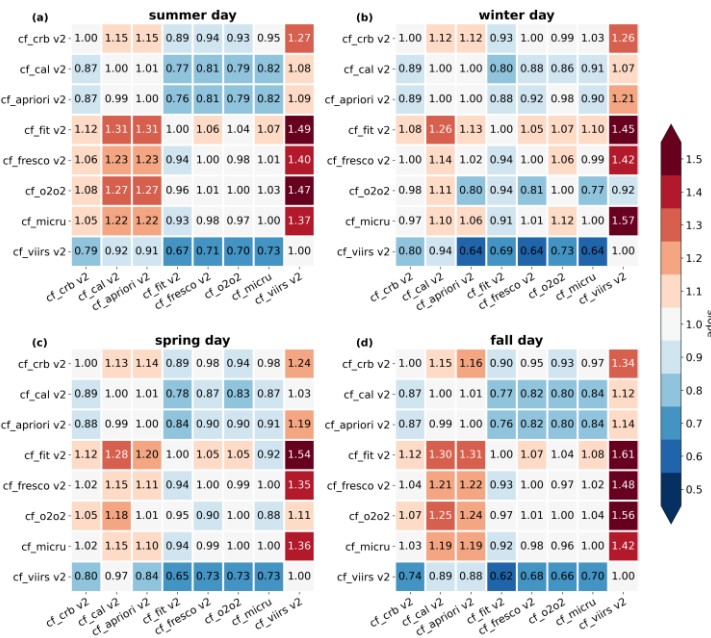

**Figure B1: Tabular intercomparison of the slopes of the scatter plots between the version 2 cloud fractions from ROCINN CRB, ROCINN CAL, OCRA a priori, the NO₂ fitting window (cf_fit), FRESCO, O2-O2, MICRU, and VIIRS for Europe and (a) the summer day, (b) the winter day, (c) the spring day, (d) the fall day.**

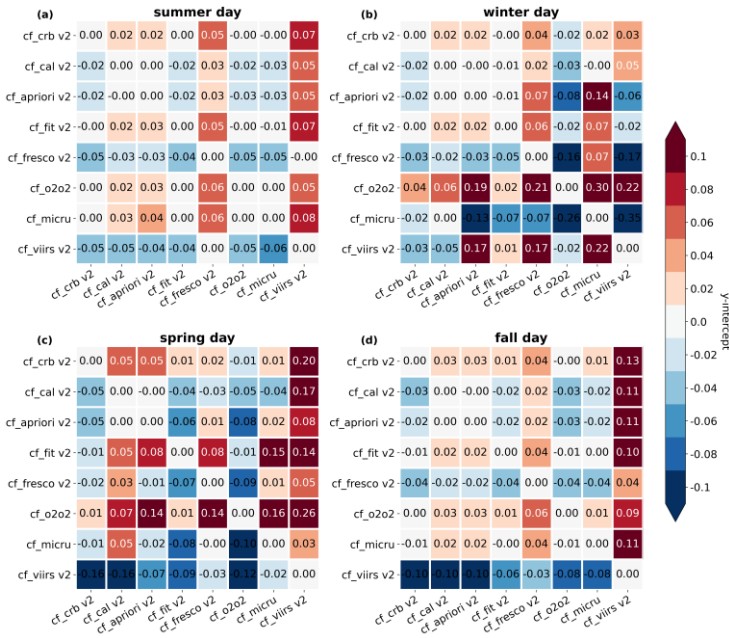

**Figure B2: Tabular intercomparison of the y-intercepts of the scatter plots between the version 2 cloud fractions from ROCINN CRB, ROCINN CAL, OCRA a priori, the NO₂ fitting window (cf_fit), FRESCO, O2-O2, MICRU, and VIIRS for Europe and (a) the summer day, (b) the winter day, (c) the spring day, (d) the fall day.**





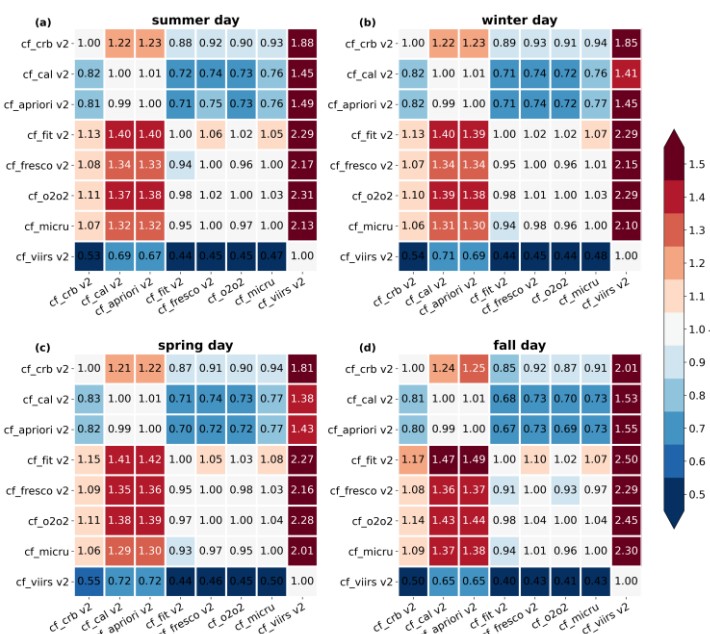

**Figure B3: As Figure B1 but for Africa.**

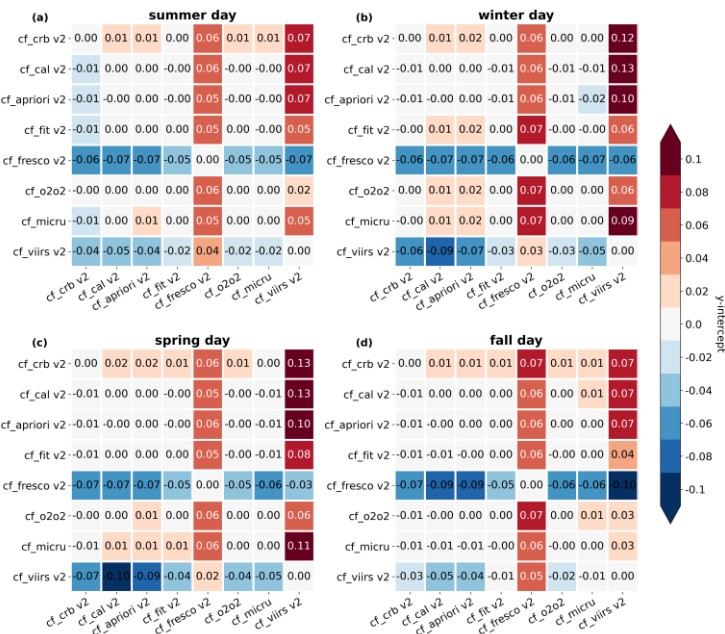

**Figure B4: As Figure B2 but for Africa.**





**Appendix C: Tabular intercomparison of the statistics between the version 2 cloud heights for China**

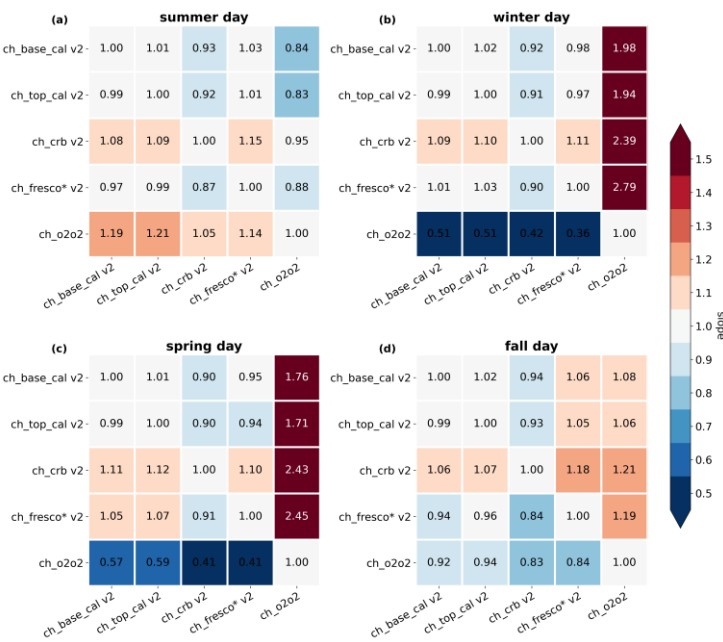

**Figure C1: Tabular intercomparison of the slopes of the scatter plots between the version 2 cloud heights from ROCINN CAL base and top, ROCINN CRB, FRESCO from the TROPOMI NO₂ product (ch_fresco*), and O2-O2 for China and (a) the summer day, (b) the winter day, (c) the spring day, (d) the fall day.**

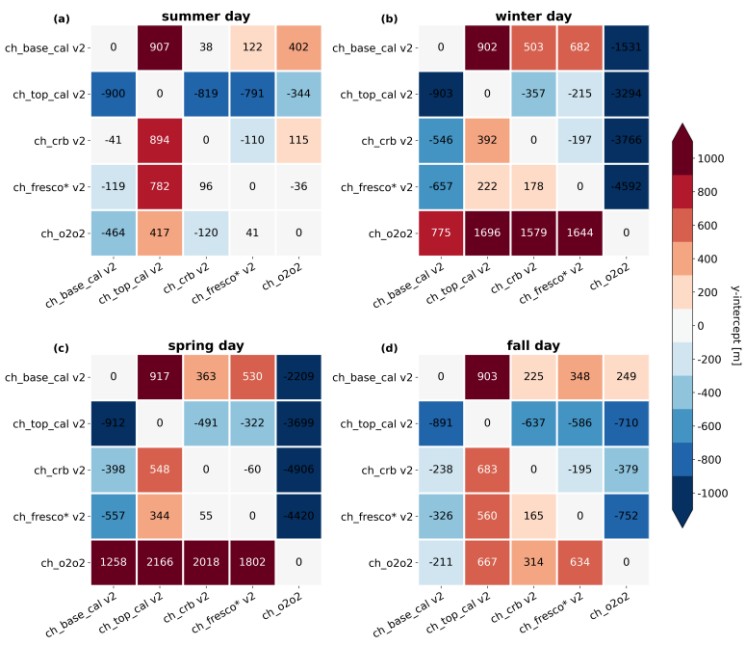

**Figure C2: Tabular intercomparison of the y-intercepts of the scatter plots between the version 2 cloud heights from ROCINN CAL base and top, ROCINN CRB, FRESCO from the TROPOMI NO₂ product (ch_fresco*), and O2-O2 for China and (a) the summer day, (b) the winter day, (c) the spring day, (d) the fall day.**



**Appendix D: Number of available cloud fraction and cloud height values after quality flagging**

The tables provide an overview of the percentage of the number of available cloud fractions (cf) and cloud heights (ch) to that of all values in the regions of Europe (Table D1), China (Table D2), and Africa (Table D3) after quality flagging with quality value of 0.5, comparing cf_fit v2 or ch_fresco* v2, respectively, with the other products. ROCINN CRB and CAL contain much fewer values compared to the other products, especially for the winter day (5 January 2019) and the spring day (4 April 2019), due to more stringent retrieval diagnostics that reduce the quality value significantly below 0.5 over snow- and ice-

covered scenes. In addition, cloud heights are only available for pixels with a non-zero cloud fraction, which explains the generally small number of available cloud height values for ROCINN CRB and CAL.

**Table D1: Overview of the percentage of the number of available cloud fraction and cloud height values to that of all values in the region of Europe after quality flagging with quality value 0.5, comparing cf_fit v2 or ch_fresco* v2 with the other products.**

| Test Day | 30 June 2018 | 5 January 2019 | 4 April 2019 | 20 September 2019 |
|---|---|---|---|---|
| cf_crb v2 | 79% | 23% | 54% | 84% |
| cf_cal v2 | 87% | 21% | 57% | 86% |
| cf_apriori v2 | 99% | 45% | 93% | 98% |
| cf_fresco v2 | 99% | 44% | 91% | 97% |
| cf_o2o2 | 99% | 45% | 93% | 98% |
| cf_micru | 99% | 45% | 93% | 98% |
| cf_viirs v2 | 99% | 45% | 93% | 98% |
| ch_crb v2 | 42% | 15% | 41% | 59% |
| ch_top_cal v2 | 50% | 13% | 44% | 61% |
| ch_base_cal v2 | 50% | 13% | 44% | 61% |
| ch_o2o2 | 99% | 45% | 93% | 98% |

**Table D2: Overview of the percentage of the number of available cloud fraction and cloud height values to that of all values in the region of China after quality flagging with quality value 0.5, comparing cf_fit v2 or ch_fresco* v2 with the other products.**

| Test Day | 30 June 2018 | 5 January 2019 | 4 April 2019 | 20 September 2019 |
|---|---|---|---|---|
| cf_crb v2 | 85% | 49% | 68% | 80% |
| cf_cal v2 | 88% | 49% | 70% | 81% |
| cf_apriori v2 | 97% | 91% | 94% | 96% |
| cf_fresco v2 | 95% | 88% | 92% | 94% |
| cf_o2o2 | 97% | 91% | 94% | 96% |
| cf_micru | 97% | 91% | 94% | 96% |
| cf_viirs v2 | 95% | 91% | 94% | 96% |





| | | | | |
|---|---|---|---|---|
| ch_crb v2 | 71% | 26% | 42% | 43% |
| ch_top_cal v2 | 74% | 25% | 44% | 44% |
| ch_base_cal v2 | 74% | 25% | 44% | 44% |
| ch_o2o2 | 95% | 91% | 94% | 96% |

**Table D3: Overview of the percentage of the number of available cloud fraction and cloud height values to that of all values in the region of Africa after quality flagging with quality value 0.5, comparing cf_fit v2 and ch_fresco\* v2 with the other products.**

| Test Day | 30 June 2018 | 5 January 2019 | 4 April 2019 | 20 September 2019 |
|---|---|---|---|---|
| cf_crb v2 | 88% | 85% | 92% | 91% |
| cf_cal v2 | 88% | 86% | 92% | 91% |
| cf_apriori v2 | 99% | 98% | 99% | 99% |
| cf_fresco v2 | 98% | 96% | 97% | 98% |
| cf_o2o2 | 99% | 98% | 99% | 99% |
| cf_micru | 99% | 98% | 99% | 99% |
| cf_viirs v2 | 99% | 97% | 98% | 99% |
| ch_crb v2 | 38% | 46% | 52% | 40% |
| ch_top_cal v2 | 38% | 48% | 52% | 41% |
| ch_base_cal v2 | 38% | 48% | 52% | 41% |
| ch_o2o2 | 99% | 97% | 98% | 99% |


**Appendix E: Across-track dependency plots of the version 2 cloud fractions for Africa and China**

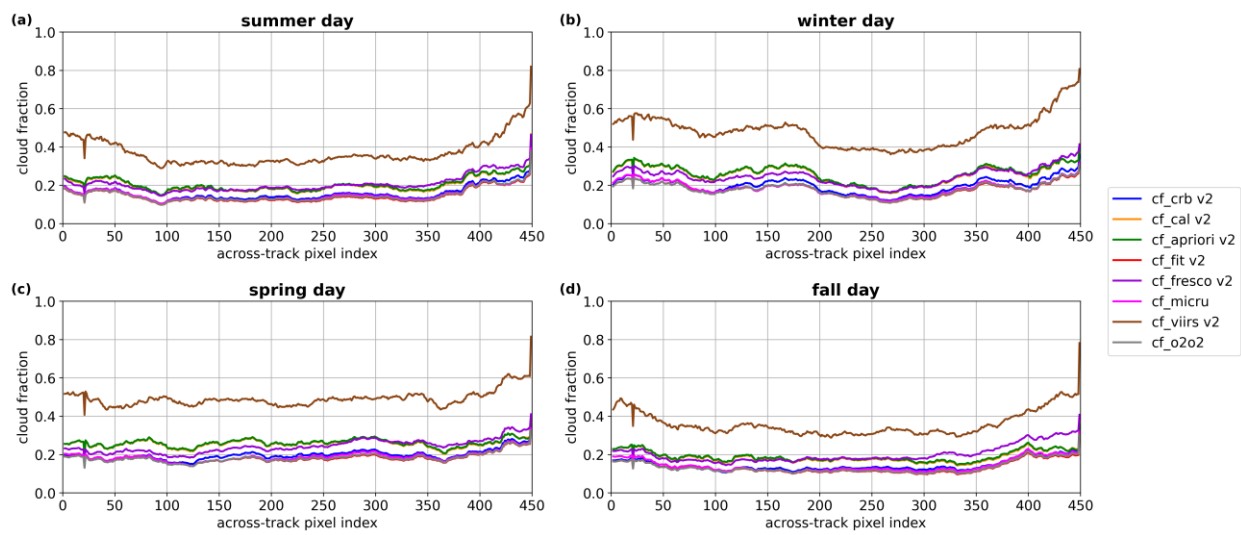

**Figure E1: Mean values of version 2 cloud fractions, as a function of the across-track index, from ROCINN CRB, ROCINN CAL, OCRA a priori, the NO₂ fitting window (cf_fit), FRESCO, MICRU, VIIRS, and O2-O2 for Africa and (a) the summer day, (b) the winter day, (c) the spring day, and (d) the fall day with quality- and snow-/ice-flagging, and including only pixels having valid values for all products.**

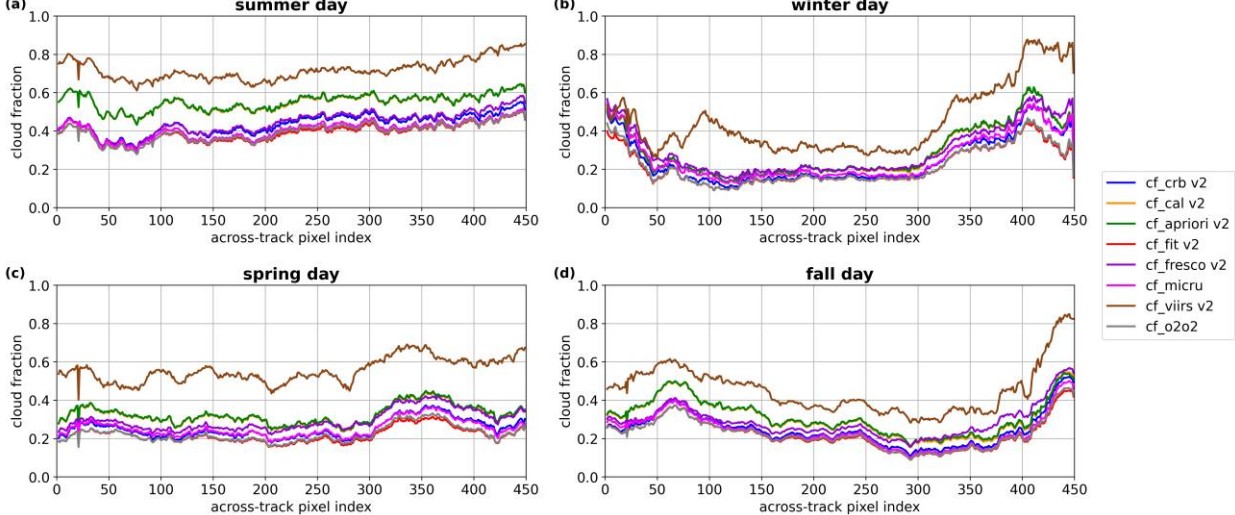

**Figure E2: As Figure E1 but for China.**



**Appendix F: Across-track dependency plots of the version 2 cloud heights for Africa and China**

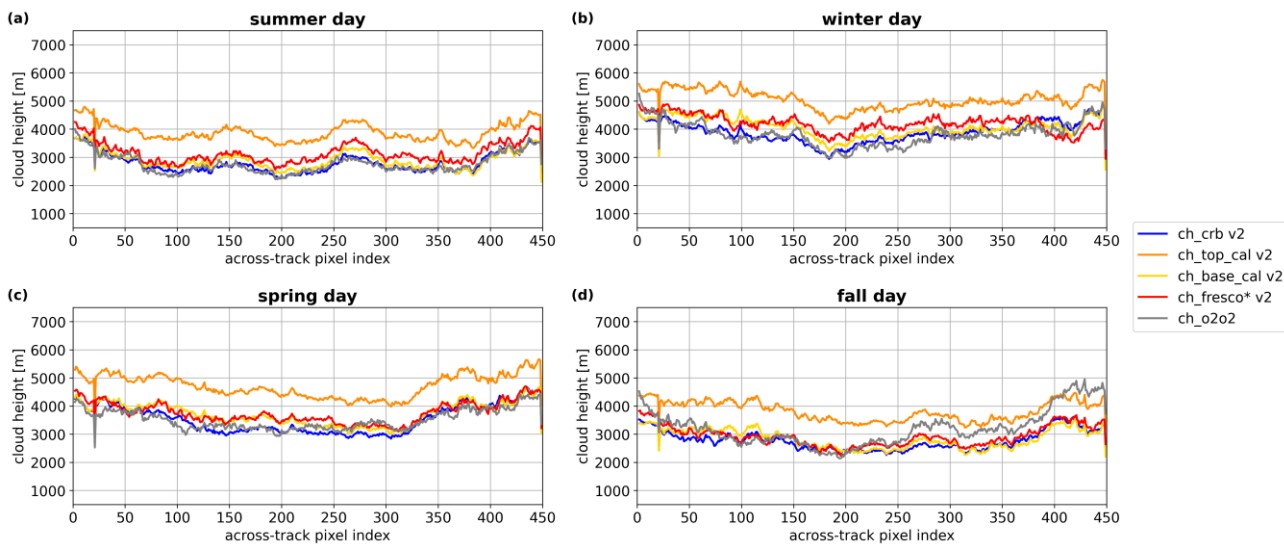


**Figure F1: Mean values of version 2 cloud heights, as a function of the across-track index, from ROCINN CRB, ROCINN CAL top and base, FRESCO from the TROPOMI NO$_2$ product (ch_fresco*), and O2-O2 for Africa and (a) the summer day, (b) the winter day, (c) the spring day, and (d) the fall day with quality- and snow-/ice-flagging, and including only pixels having valid values for all products.**


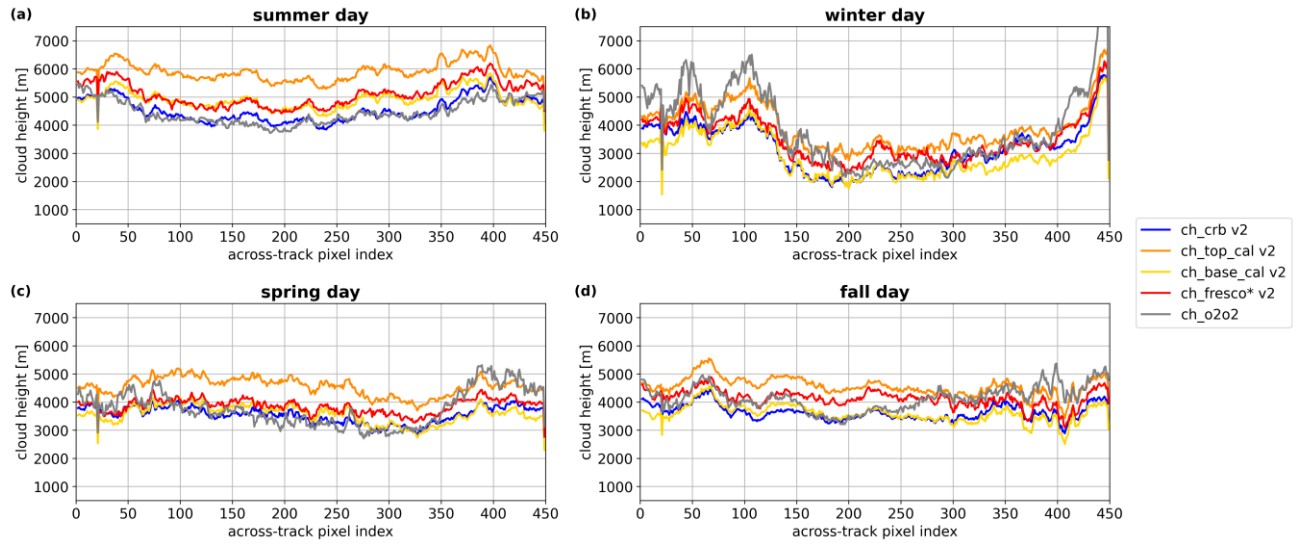

**Figure F2: As Figure F1 but for China.**



*Data availability.* TROPOMI L2 CLOUD (OCRA/ROCINN), FRESCO, NO$_2$, and NP_BD3 (VIIRS) data from July 2018
onwards is publicly available via the Sentinel-5 Pre-Operations Data Hub (https://s5phub.copernicus.eu/). MICRU data was
provided by the Max-Planck-Institute of Chemistry (MPIC) Mainz, Germany, and O2-O2 data was provided by the Royal
Netherlands Meteorological Institute (KNMI).

*Author contributions.* ML and AR designed the study. ML performed the data analysis and wrote the manuscript with
contributions from AR, JPB, TW, HE, NT, RL, and DL. HS made the MICRU data available, and MS made the O2-O2 data
available.

*Competing interests.* Some authors are members of the editorial board of journal AMT. The peer-review process was guided
by an independent editor, and the authors have also no other competing interests to declare.

*Acknowledgements.* Parts of this work were funded by the European Space Agency (ESA) through the S5P/TROPOMI Mission
Performance Centre (S5P-MPC) under the ESA contract No. 4000117151/16/1-LG ("Preparation and Operations of the
Mission Performance Centre (MPC) for the Copernicus Sentinel-S Precursor Satellite"), the Deutsches Zentrum für Luft- und
Raumfahrt (DLR) under contract 50EE1811A ("S5P Datennutzung"), and the University of Bremen.





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
