# Peer review of "Intercomparison of Sentinel-5P TROPOMI cloud products for tropospheric trace gas retrievals"

_Atmospheric Measurement Techniques, 2022_

## Author Comment (AC1)

**AMT-2022-122 Author's comment to RC1 by Referee #3 (Miriam Latsch et al.)**

Cloud parameters (cloud fraction and cloud height) from different cloud retrieval algorithms for Sentinel-5 Precursor (S5P) TROPOMI are compared in scatter diagrams, latitude-longitude maps, tabular intercomparisons, and daily across-track intercomparisons. The variety of graphs of the intercomparisons are insightful and allows the reader to assess adequately the general aspects of the intercomparisons in a quantitative manner.

We thank Referee #3 for the comments. Detailed responses to the comments can be found below.

Main points:

This paper compares the different cloud products such as cloud fraction and cloud height, but goes no further than the intercomparisons. The intercomparisons need to be placed into context to the uncertainties of the primary recommended cloud fraction and cloud height TROPOMI data products. Are particular cloud products recommended by the TROPOMI science team? If so, which data products? If not, this should be stated. What are the cloud fraction and cloud height uncertainties of the primary products (if recommended), and how do these uncertainties relate to the spread in the intercomparisons presented in this paper?

A recommendation for a specific cloud product is not the subject of this paper and would also not make sense as each application has its own requirements for the optimum cloud product. Therefore, we aim at giving an overview of the existing TROPOMI cloud products and highlight their differences but are not judging which product is best to use.

The formal mission product requirements for the CLOUD (OCRA/ROCINN) retrieval algorithm state that the uncertainty due to systematic effects is 20% for cloud fraction, cloud height, cloud albedo and cloud optical thickness, and the uncertainty due to random effects is 0.05 for cloud fraction and cloud albedo, 0.5 km for cloud height, and 10 for cloud optical thickness (from the recent validation report ROCVR by Lambert et al., 2022).

In order to address the point of the reviewer, we renamed the Section 4 to "Summary and Conclusions" and added a paragraph in Line 639 to give a better overview and summary about the TROPOMI cloud products:
"Several different cloud products are included in the S5p operational lv2 data, and they differ in their definition and typical application. The OCRA a priori cloud fraction operates in the UV/VIS spectral region and is used as input for the ROCINN CRB and CAL models, which operate in the NIR spectral region. For global statistics, using the cloud fraction from OCRA or from the ROCINN products will not make a big difference. For individual measurements, particularly over snow and ice cover, it is recommended to use the ROCINN CRB and CAL cloud fractions instead of the OCRA a priori cloud fraction. FRESCO provides an effective cloud fraction retrieved from top-of-atmosphere reflectances, assuming an optically thick Lambertian cloud with a fixed albedo of 0.8. This approach is useful for trace gas retrievals, e.g., for ozone. FRESCO retrieved in the $O_2$ A-band becomes sensitive to systematic uncertainties of the surface albedo with strong viewing angle dependence, especially over forests. Due to the large difference between

the $O_2$ A-band and the $NO_2$ retrieval window and the misalignment between the TROPOMI ground pixel view of the VIS and NIR bands, the cloud fraction from the TROPOMI $NO_2$ product has been developed. It is suitable for $NO_2$ trace gas retrievals because the cloud fraction is retrieved from the $NO_2$ fitting window in the UV/VIS spectral region at 440 nm. The O2-O2 algorithm uses measurements from the O2-O2 ($O_4$) absorption window at 477 nm and assumes a fixed cloud albedo of 0.8. Although a similar model as the one in FRESCO is used, the O2-O2 cloud product is more sensitive to lower clouds and to aerosols due to the application of O2-O2 collision-induced absorption. It also provides continuity with data from the OMI mission. The MICRU algorithm is optimized for low cloud fractions smaller than 0.2 and is preferred for pixels over water with sun glint due to the explicit treatment of sun glint. The VIIRS cloud fraction is a geometric cloud fraction retrieved from a 4-level cloud mask with cloud probability. It does not depend on cloud optical thickness as strongly as an effective cloud fraction and thus, it shows a good performance for selecting completely cloud-free scenes. Therefore, it is useable for cloud screening by TROPOMI products, e.g., the methane processor, to identify cloud free scenes for processing."

As an additional request, Figure 16 cloud fraction curves of various algorithms have a spread of 0.2 in the cloud fraction values. Other instruments also generate cloud fractions. What are the uncertainties associated with e.g. MODIS, and how do the MODIS uncertainties compare to the TROPOMI spread in the cloud products? This sort of additional information will enhance the value of the paper. It is not requested to do MODIS – TROPOMI data intercomparisons, but a several- sentence discussion of typical MODIS cloud data product uncertainties would be informative.

The reviewer suggests a comparison of the spread of TROPOMI cloud fractions with uncertainties from MODIS cloud fractions. This would make sense if cloud fractions were universal and well-defined quantities. However, as discussed in the manuscript, the various cloud fractions are based on different assumptions and definitions, and therefore vary in their values. This is not the same as the uncertainty of the values from for example measurement noise, shadowing, or uncertainties in the surface reflectance used which all add additional variability to the data. We therefore do not see the benefit of comparing the differences between different cloud fraction types to uncertainties of MODIS cloud products.

Several paragraphs need to be added to the paper to address these requests before publication.

Minor points

My copy of the paper does not have indented paragraphs. Should line 49 and subsequent paragraphs be indented?

The formatting of the preprint followed the template provided by AMT. The final manuscript will be typeset by the journal and will adhere to all style rules.

The Figure 5 and Figure 11 numbers and x axis labels associated with the small boxes are too small to be readable. Please increase the font size.

The numbers and the axis labels of these plots in the manuscript and the supplement have been enlarged.

Line 586 has a blank line. Please correct this typo.

This has been corrected.

Criteria

1. Does the paper address relevant scientific questions within the scope of AMT? Yes, intercomparisons of data products fall within the scope of AMT.

2. Does the paper present novel concepts, ideas, tools, or data? The intercomparisons are useful and straightforward. Novel concepts are not introduced by the paper.

3. Are substantial conclusions reached? The intended outcomes of the intercomparisons are discussed in an adequate manner.

4. Are the scientific methods and assumptions valid and clearly outlined?  Yes

5. Are the results sufficient to support the interpretations and conclusions? Yes

6. Is the description of experiments and calculations sufficiently complete and precise to allow their reproduction by fellow scientists (traceability of results)? Yes

7. Do the authors give proper credit to related work and clearly indicate their own new/original contribution? Yes

8. Does the title clearly reflect the contents of the paper? Yes

9. Does the abstract provide a concise and complete summary? Yes

10. Is the overall presentation well-structured and clear? Yes

11. Is the language fluent and precise? Yes.

12. Are mathematical formulae, symbols, abbreviations, and units correctly defined and used? Yes.

13. Should any parts of the paper (text, formulae, figures, tables) be clarified, reduced, combined, or eliminated? The intercomparisons need to placed into context (related to) the uncertainties of the primary TROPOMI and/or other instrument (e.g. MODIS) cloud data products used by the research community. See Main Points above.

14. Are the number and quality of references appropriate? Referencing of the various algorithms is adequate.

15. Is the amount and quality of supplementary material appropriate? Yes

Thank you for this overview and your feedback.

**References**

Lambert, J.-C., Keppens A., Compernolle S., Eichmann K.-U., de Graaf M., Hubert D., Langerock B., Ludewig A., Sha M. K., Verhoelst T., Wagner T., Ahn C., Argyrouli A., Balis D., Chan K. L., De Smedt I., Eskes H., Fjæraa A. M., Garane K., Gleason J. F., Goutail F., Granville J., Hedelt P., Heue K.-P., Jaross G., Kleipool Q., Koukouli M.-E., Lorente Delgado A., Lutz R., Michailidis K., Nanda S., Niemeijer S., Pazmiño A., Pinardi G., Pommereau J.-P., Richter A., Rozemeijer N., Sneep M., Stein Zweers D., Theys N., Tilstra G., Torres O., Valks P., van Geffen J., Vigouroux C., Wang P., and Weber M.: Quarterly Validation Report of the Copernicus Sentinel-5 Precursor Operational Data Products #15: April 2018 – May 2022, S5P MPC Routine Operations Consolidated Validation Report series, S5P-MPC-IASB-ROCVR-15.01.00-20220713, issue #15, version 15.01.00, 212 pp., https://s5p-mpc-vdaf.aeronomie.be/ProjectDir/reports//pdf/S5P-MPC-IASB-ROCVR-15.01.00_20220713_signed.pdf, 13 July 2022.

---

## Author Comment (AC2)

**AMT-2022-122 Author's comment to RC2 by Referee #4 (Miriam Latsch et al.)**

This manuscript shows a large variety of descriptive statistical comparisons between mutiple cloud products from the TROPOMI instrument on S5P and VIIRS on SNPP. The information is thorough in scope (though limited in time, as only 4 days of data are used), and is valuable to the remote sensing and applications communities that use S5P data. The thoroughness of the comparisons means there is a great number of figures, and in some placed the organization is tough to follow, and I have made suggestions below. There are also a number of places where the discussion and analysis is unclear or misleading. I do think the manuscript is publishable but in the present state I would recommend a large number of changes.

We thank Referee #4 for the feedback and the suggestions. The detailed responses to the comments can be found below. The line numbers refer to those in the preprint version.

Overall major comments:

It is unclear if the VIIRS cloud mask should even be included in the manuscript. Whenever the VIIRS product appears in a figure, the correlation with other products is poor, and the reader is reminded "the geometric VIIRS cloud fraction is expected to have the largest differences...". Since it is a fundamentally different product, why is it included in this comparison? Are there cases where the VIIRS cloud mask should be considered from a user's perspective?

The VIIRS cloud fraction is determined from another platform, the Suomi NPP satellite, but mapped to the TROPOMI footprint. The VIIRS product is used by different TROPOMI products for cloud screening: The methane ($CH_4$) processor uses it to identify cloud-free scenes for processing, for the CLOUD processor it is used for cloud screening in the daily surface albedo retrieval (GE_LER), and it is used as a cloud mask for the aerosol layer height (ALH) processor. Therefore, it is interesting for S5p users to see how it relates to the products that can be derived from TROPOMI, which is now uniquely demonstrated in this paper.

To motivate the inclusion of VIIRS data, in Line 186, the following sentence has been added:
"The VIIRS cloud product is used for cloud screening by different TROPOMI products, e.g., by the methane ($CH_4$) processor to identify cloud-free scenes for processing, the CLOUD processor for cloud screening in its daily surface albedo retrieval (GE_LER), and the aerosol layer height (ALH) processor as a cloud mask."

It is unclear why the region over China was selected, particularly if the "values for Europe ... and China .. behave very similar" (Line 344). Could the China regional subset simply be excluded from the paper entirely? (Which would substantially reduce the number of figures). I would recommend moving the plots related to the China regional subset to the Supplement only, rather than switching regions in the main manuscript, with no justification (e.g., Europe and Africa are used for all Cloud fraction comparisons, but then the

presentation switches to China for Cloud height. Why was this done? It would be clearer to do the comparisons for only the Europe subset in the main paper.)

China is the region with the highest air pollution on Earth, so we would like to include it in our comparison. We thought it would be useful to show results for only one region per cloud parameter to reduce the number of figures in the manuscript. We decided to show Europe because there we can find snow and ice cover, Africa for the sun glint issue due to the water surfaces included, and China for the cloud heights because this is the most pollutant region, representing a challenge for the cloud retrieval algorithms to retrieve the cloud height accurately.

In Line 248, the following sentences are added to give an explanation why the three regions were chosen to compare the cloud products in this paper:
"The different regions were selected to investigate the performance of the cloud retrieval algorithms in different challenging situations: The Europe region includes snow and ice cover, the Africa region is suitable for sun glint due to the water surfaces around the continent, and the China region represents the most polluted region on Earth, which poses a challenge for the cloud products to accurately retrieve cloud heights."

On the slopes and intercepts (appearing first in Figure 2 and many figures thereafter): for many of the comparisons, there are subpopulations in different regions, which makes the single linear fit questionable. For example, figure 7F is one of the worst examples: for the main group of points, the intercept is nearly zero and the slope is slightly larger than 1, whereas the single linear fit shows a large intercept and slope less than 1. Therefore, many of the differences between algorithms for the slopes & intercepts are of similar size to biases in the slopes and intercepts due to the data distributions themselves. I would recommend removing the fits entirely, and removing the associated "Tabular intercomparison" figures (B1, B2, etc.)

We agree with the reviewer that the results of the linear regression are often strongly affected by subgroups of the points. Following the suggestion of the reviewer, the tabular intercomparison figures of the slopes and the y-intercepts have been removed from the appendix and the supplement, as well as the references to the Appendix B and C. The regression line itself was not removed from the density histogram plots, since it is an important statistical tool to find out how the subpopulations count to the distribution.

In line 535: the authors state, "...relatively small data samples may also have across-track variations .." I strongly agree with this statement, and I think this calls into question the use of the regional data subsets for the purpose of examining across-track biases. All of the cross-track plots of regional subsets appear too noisy to be of any use, and I recommand removing them from the manuscript. The plots could be retained in the supplement, but I would not include detailed discussion or analysis of features seen in the regional subset plots.
For the cross-track plots, the differences between algorithms generally quite small and subtle (other than the results from the VIIRS algorithm), and the plots are very hard to

interpret. I cannot discern the color difference between many of the lines: it would be better choose a smaller subset of more clearly separate colors (perhaps 4) and use solid/dashed line styles in addition. The data would also be more clearly displayed with the reference algorithm "cf_fit" displayed as cloud fraction, and then the other algorithm data could be displayed as differences in CF relative to cf_fit. (and equivalent for the CH: plot ch_fresco and then the differences relative to ch_fresco.)

Following the suggestion of the reviewer, the plots of the regional across-track dependency and their discussion for the cloud fractions and the cloud heights have been removed from the manuscript. The former can now be found in Section S3 of the supplement. The text of Section 3.2.5 and the conclusions have been adjusted for the cloud fraction and cloud height comparisons, now only discussing the global plot.

In these plots, we show the across-track dependency of each cloud product separately and not in comparison to cf_fit/ch_fresco*, as there is no reason to assume that the across-track variability of this product is the correct one. To better separate the different cloud products, the colours of the lines for the cloud fractions and the cloud heights were rearranged so that similar colours are no longer close to each other (in the manuscript and supplement). Furthermore, the scale of the y-axis of the global plots in the manuscript has been reduced to stretch the offsets between the cloud products and make them easier to distinguish.

There is lack of discussion at the end of the manuscript for how a potential S5P data user should interpret all this information. The main conclusion seems to be "things are more in agreement, but there are still a lot of differences, and quality flagging matters" - which is not particularly useful for a user. Can the authors provide some sort of guidance here, or would that be considered 'out of scope'?

A recommendation for a specific cloud product is not the subject of this paper and would also not make sense as each application has its own requirements for the optimum cloud product. Therefore, we aim at giving an overview of the existing TROPOMI cloud products and highlight their differences but are not judging which product is best to use.

In order to address the point of the reviewer, we renamed the Section 4 to "Summary and Conclusions" and added a paragraph in Line 639 to give a better overview and summary about the TROPOMI cloud products:
"Several different cloud products are included in the S5p operational lv2 data, and they differ in their definition and typical application. The OCRA a priori cloud fraction operates in the UV/VIS spectral region and is used as input for the ROCINN CRB and CAL models, which operate in the NIR spectral region. For global statistics, using the cloud fraction from OCRA or from the ROCINN products will not make a big difference. For individual measurements, particularly over snow and ice cover, it is recommended to use the ROCINN CRB and CAL cloud fractions instead of the OCRA a priori cloud fraction. FRESCO provides an effective cloud fraction retrieved from top-of-atmosphere reflectances, assuming an optically thick Lambertian cloud with a fixed albedo of 0.8. This approach is useful for trace gas retrievals, e.g., for ozone. FRESCO retrieved in the $O_2$ A-band becomes sensitive to systematic uncertainties of the surface albedo with strong

viewing angle dependence, especially over forests. Due to the large difference between the $O_2$ A-band and the $NO_2$ retrieval window and the misalignment between the TROPOMI ground pixel view of the VIS and NIR bands, the cloud fraction from the TROPOMI $NO_2$ product has been developed. It is suitable for $NO_2$ trace gas retrievals because the cloud fraction is retrieved from the $NO_2$ fitting window in the UV/VIS spectral region at 440 nm. The O2-O2 algorithm uses measurements from the O2-O2 ($O_4$) absorption window at 477 nm and assumes a fixed cloud albedo of 0.8. Although a similar model as the one in FRESCO is used, the O2-O2 cloud product is more sensitive to lower clouds and to aerosols due to the application of O2-O2 collision-induced absorption. It also provides continuity with data from the OMI mission. The MICRU algorithm is optimized for low cloud fractions smaller than 0.2 and is preferred for pixels over water with sun glint due to the explicit treatment of sun glint. The VIIRS cloud fraction is a geometric cloud fraction retrieved from a 4-level cloud mask with cloud probability. It does not depend on cloud optical thickness as strongly as an effective cloud fraction and thus, it shows a good performance for selecting completely cloud-free scenes. Therefore, it is useable for cloud screening by TROPOMI products, e.g., the methane processor, to identify cloud free scenes for processing."

The organization of the supplemental figures is tough to follow. It would help to have a table of contents or similar at the beginning of the supplement document. It also would be helpful to group all the version 1 plots into a separate section. Many readers will not care about the old products: it would be more useful to be able to skip over those plots entirely.

Thank you for this comment. The supplement has been reorganized so that the version 1 plots are now at the end of the supplement in a separate section. A table of contents has also been added. The references in the manuscript text to the supplement have been adjusted to the new order.

Specific minor comments:

There are multiple places where the explanation is very jargon heavy, specific to terms related to S5P. (some are noted below)

The "OCRA" algorithm is described as the cloud fraction a priori. Clarify which algorithms actually use this as their a priori - is it just ROCINN, or do others? If OCRA is intended as an a priori, does it make sense to even include in this comparison? Shouldn't the algorithm that uses OCRA as the a priori be an improvement over the OCRA result?

ROCINN (operating in the NIR) is the only algorithm that uses the cloud fraction a priori from OCRA (operating in the UV/VIS). The main retrieval parameters from ROCINN are the cloud top height and cloud optical thickness (in ROCINN CAL) and the cloud height and cloud albedo (in ROCINN CRB). In addition to those two main retrieval parameters, the cloud fraction is also a retrieval parameter in ROCINN where it is strongly regularized against the input a priori cloud fraction from OCRA. Due to the strong regularization of the a priori, the ROCINN cloud fractions in the CAL and CRB model differ not too much from

the input a priori from OCRA. In certain scenes (e.g., OCRA a priori cloud fraction overestimation over very bright surfaces) the ROCINN cloud fraction retrievals may indeed improve the a priori by correcting the overestimation. In the TROPOMI L2 cloud product, three cloud fractions are provided: the a priori cloud fraction from OCRA, the cloud fraction from the ROCINN CRB model and the cloud fraction from the ROCINN CAL model. This is the reason why the OCRA a priori cloud fraction is included in the comparisons of this paper, because it is a part of the TROPOMI L2 cloud product. For global statistics, using one or the other cloud fraction will not make a big difference. For individual measurements (particularly over snow/ice) it is recommended to use the cloud fraction from the ROCINN retrieval.

Line 216/Equation (2) - this is missing a minus sign? or should be (1 - (CP/Ps)...)?

Thank you for this hint! The minus sign was inserted in Equation (2).

Line 229 - 233: this is very unclear. Why does FRESCO have a different number of crosstrack pixels? Table 1 says that FRESCO, ROCINN and MIXCRU all use the NIR wavelengths.

FRESCO is the only algorithm retrieving cloud fractions from the NIR channel (band 6 – BD6), and not from the UV/VIS channel (band 3 – BD3) like the other cloud products. The number of ground pixels is different because the NIR detector has a slightly different spatial coverage (footprint) than the UV/VIS detector and therefore, no full overlap is achieved.

Line 236: how many VIIRS cloud mask pixels fall within one TROPOMI field of view, on average?

The following sentences were added to Line 182:
"It flies in a so-called loose formation with S5p, with the difference in overpass time being less than 5 minutes."
and to Line 184:
"The number of VIIRS cloud mask pixels within a given S5P scene is typically about 50-200, depending on cross-track position for S5P VIS/NIR bands and about double that in the SWIR."

Line 239: These all look like density histograms, not 'scatter plots'. (these are matplotlib hexbin plots, correct? then they are histograms, not scatter plots)

Thank you for pointing this out – we have corrected the term "scatter plots" to "density histograms" in all figure captions and the manuscript text.

Line 279: what is "BD3"? (S5P jargon, I assume?)

This abbreviation was already mentioned in Line 135. What is meant is band 3 (BD3), which is the UV/VIS band, while BD6 (band 6) is the NIR band.

We changed the sentence in Line 135 from "…and the co-registration between the UV/VIS (BD3) and the NIR (BD6) bands is improved." to "…and the co-registration between the UV/VIS band 3 (BD3) and the NIR band 6 (BD6) is improved.".

Line 280: instead of "fixed pixel shift", do you mean "integer pixel shift"?

Yes, "integer pixel shift" is correct. The term in Line 280 has been changed and the sentence in Line 282 has been adjusted to:
"However, while version 1 used a simplified integer pixel shift, in version 2, an improved scheme using the TROPOMI static mapping tables provided by KNMI was implemented. These mapping tables contain the actual overlap areas of neighbouring pixels and are therefore much more precise. Due to the co-registration, it may happen, e.g., along cloud edges, that pixels with cloud fractions larger than 0.05 in one band may have cloud fractions smaller than 0.05 in the other band. Since a 0.05 cloud fraction is the threshold for the ROCINN retrieval, this might result in slightly different data yields in the two bands."

Line 285/Figure 2 (and other figures): it is confusing that the algorithm names are not consistent between the text and the plot annotations. (specifcally, "ROCINN" and "OCRA" are not used in the figures. "NO2" became "fit")
For Figure 2, and all similar figures: the colormap appears to be deeply saturated in all cases (meaning, there are many grid cells with values larger than the highest value in the color bar.) These colormaps should be switched to a logarithmic scale.

The cloud products and their abbreviations for the plots are explained in detail in Table 1. Our intent was to keep the names as short as possible. The name "$NO_2$" instead of "fit" may be confusing to the reader because the cloud fraction cf_fit is retrieved from the $NO_2$ fitting window ("fit") but can be used not only as a cloud product for the $NO_2$ trace gas but also for other purposes.

We have tried in the past to use a logarithmic scale and felt that this was not helpful for an accurate representation of the results, therefore we prefer to accept the oversaturation as a part of the results.

Line 291: how could a degradation correction create the subpopulations in figure 2e? Shouldn't a degradation correction apply to all points, making the slight less-than-1 slope seen in 2e?

We have mentioned two contributions: handling radiance levels that would lead to a cloud fraction greater than 1, and the irradiance degradation correction. The latter will indeed not lead to a split of populations, like the one we see in Figure 2e. Another contribution is responsible for this: the data sources for the snow ice information have been switched from NISE to ECMWF, as explained in Lines 291-296. This changes the sea-ice fraction and the land-snow flag, and through that, the surface albedo for some pixels as the

snow/ice information is used to modify the surface albedo. This in turn has an impact on the cloud fraction. Pixels in version 1 that were previously snow/ice-free may now be classified as containing snow/ice in version 2 and vice versa, causing extra branches in figure 2e. We think the new situation better represents the actual scene.

Line 300 caption: Typo, this says "v1 - v1", should be "v2 - v1"?

The label of the colour bar of the difference plot has been corrected to "cf_fit v1 minus cf_fit v2".

Line 312: would suggest saying "scatter" (the spread of values in the graph) instead of "scattering" (physical process of light interacting with particles).

Yes, that sounds better! We changed it in Line 312, as well as in Lines 427, 449, 503, 518, and 696.

Line 315: sentence "In addition, the error in the FRESCO cloud height..." is very unclear.

The sentence was removed.

Line 317: I do not see how you can argue that the FRESCO matches ROCINN better by what is presented in Figure 4. We are only presented with comparison between each algorithms' version 1 and version 2, not between the algorithms.

We do not say that they are in better agreement, only that the FRESCO cloud height in version 2 is larger than that in version 1 in some cases. Because the FRESCO cloud height is smaller in version 1 (Compernolle et al., 2021), our results may indicate that the FRESCO values are now closer to the ROCINN cloud heights. However, in Line 319, the sentence "This assumption is corroborated in Section 3.2.5." is added to clarify that this statement is only an assumption. In Section 3.2.5., this is analysed in more detail when comparing the across-track plots, with the following sentence being added in Line 593: "This is the major difference from version 1, where ch_fresco* is more consistent with or even smaller than ROCINN CRB and CAL CBH (Section S4.3)."

FIgure 5, and other later figures: the subplot labels "summer day", "winter day", etc, are ambiguous. Replace these with the actual dates used.

The subplot labels of the tabular intercomparison plots and the across-track plots have been changed to the actual dates used.

Line 380-382: unclear, what is 'polynomial and Ring term'? more jargon?
"This leads to a bit higher values for cf_fit than for O2-O2" - I see almost no data points in figure 7e, with cf_fit larger than O2-O2.

The bracket "(polynomial and Ring term)" in Line 382 has been removed.

The correction for the $NO_2$ absorption is most important for small cloud fractions. Thus, the argumentation is consistent with our results because cf_fit is larger than O2-O2 for small cloud fractions in Figure 7e. The sentence in Lines 382-383 has been changed from "This leads to a bit higher values for cf_fit than for O2-O2, since the trace gases absorb some of the light." to "This leads to a bit higher values for cf_fit than for O2-O2 for small cloud fractions, where the correction for the $NO_2$ absorption is especially of importance, since the trace gases absorb some of the light.".

Also, the fact that cf_fit is on average smaller than O2-O2 is explained in more detail, and the following sentences were added to Line 384, where the second difference between cf_fit and O2-O2 is commented (the albedo is evaluated at a different wavelength): "This leads to some small changes at larger values. Another reason why cf_fit is on average smaller than O2-O2 at the highest values is that the lookup table used by both cloud retrieval algorithms has a weak dependence on the cloud height, which is different in both cases (FRESCO and O2-O2 cloud heights). These heights may differ substantially, depending on the location, which influences the total radiance level. This is a difference which may lead to subtle changes in cloud fraction because if the cloud goes up, the Rayleigh scattering decreases, the intensity predicted by the cloud lookup table decreases, and the computed cloud fraction increases."

Line 400: figures 8d and 8f show negative differences that are spatially correlated to the snow-ice mask from Figure 3, but 8c and 8g show negative differences over much different spatial domains. In fact the large region centered at 50E, 60N, in 8g is snow covered but the CF values are nearly identical.
In figure 8 the display range appears to be too narrow, in particular 8g appears to be mostly out of range. A nonlinear level spacing might help - perhaps [0.1, 0.3, 0.6] instead of [0.1, 0.2, 0.3] for the major levels in the colorbar?

The oversaturation shows that the cloud products in Figure 8c and 8g strongly differ from cf_fit and also from the other cloud products because they do not have such large difference values compared to cf_fit. Therefore, adjusting the colour bar will not help our study to show more results because the smaller differences will then become weaker. Furthermore, the majority of pixels in Figure 8g have a difference of 1.0, as can also be seen in Figure 7g (density histogram). Therefore, an adjustment of the colour bar will not eliminate the oversaturation.

Figure 10 & Line 447 - Line 452, it is concluded that MICRU is doing better over the ocean glint, but this is not demonstrated by the difference plot. From the difference plot, we only know that the two algorithms respond differently in the glint region, not which is more correct. It would help to add another panel that shows the actual cloud fraction reference map (from cf_fit), and there is room in the upper right. Presumably, the cf_fit should show a false excess cloud fraction that lines up to the glint region shapes we see in the difference plot (8f). The color map is again saturated (same as Fig. 8)

Good point. In order to address it, a cloud fraction map has been added to the difference plots of the region Africa to show the apparent cloudiness in the cf_fit map matching the sun glint structure. In Line 449, the reference to "Figure 10d" is added to point to the new panel with the cloud fraction map of cf_fit. In addition, in Line 448, "Figure 10f" is changed to "Figure 10g". The caption of Figure 10 is adjusted, and the following sentence is added: "Panel (d) shows the cloud fraction map of cf_fit over the Africa region for that day."
The cloud fraction maps of cf_fit for the region of Africa were also included in Figure S29, S31, and S33 in the supplement.

Figure 13 - same comment as Figure 8, 10, the color scale is heavily saturated. A nonlinear scale should help.

The colour scale of the cloud height difference plots has been extended (change of maximum from 1000 m to 3000 m).

Figure 15: why does panel (d) show such good aggrement compared to 14d? I think Fig 15 should have the same data as 14, just "zoomed-in"? Is this a trick of the way the data is 'saturating' the color scale? (see earlier comment)

Figure 15 is only a zoomed-in of Figure 14, but due to the removal of the larger cloud heights, the density points only represent the percentage of the cloud heights below 2 km, although the colour scale has the same range. In addition, due to the lower height range, the size of the hexbins in Figure 15 is much smaller compared to Figure 14. Thus, this is just an effect of the presentation of the data and the distribution in Figure 15d is not really better than that in Figure 14d.

Line 530: "No observational effects for full cloud cover ..." - I do not think this is true. For fully cloudy pixels there are still anisotropies in the scattered radiation, that will conflict with the Lambertian cloud models used in many of the retrievals.

You are right. For FRESCO and O2-O2, a Lambertian cloud is used without taking into account any anisotropy that is potentially caused by scattering on cloud particles. The sentence "No observational effects …" in Line 530 has been removed from the paper.

Line 560 "... very similar run of the curves ..." what does this mean?

The following addition is made to the sentence in Line 560: "…, consistent with the good agreement in the density histograms."

Line 573-577: if the "dip" is due to the change in the binning scheme, shouldn't there be a similar feature on the other side of the cross track scan?

Due to changes in the binning scheme, the footprint areas decrease close to the western and eastern edges of the swath. This may cause small dips or peaks in the across-track

dependency plots at the transition region due to the co-registration scheme. At the western transition, three pixels contribute to the co-registration, while at the eastern transition only two pixels contribute to the co-registration. This might be a reason for a more pronounced effect at the western transition region. The impact also seems to depend on the parameter considered.

Line 580 "no overall indication for a systematic across-track problem ... except for the above-mentioned FRESCO issue over Africa."
I do not understand this claim, nor the FRESCO issue. In Figure E1, isn't the green line (cf_apriori) the outlier here (ignoring VIIRS)?
Note that these claims are relative to the regional subset data, where the data is too noisy to make these conclusions (see earlier major comment) - in any of one of the subsets the actual mixture of clouds observed across track could be very different. If one algorithm tends to have a bias for low clouds but not high clouds, this would then manifest as a cross-track bias.

The regional across-track dependence plots and their discussions have been removed from the manuscript (see response to the earlier major comment).

Line 595: doesn't the yellow line (ch_base_cal) also show a 'step' due to the interpolation (is this the feature at cross track pixel 22?) If not, I am not sure what 'steps' are notable in Figure 19.

By steps is not meant the dip at across-track pixel 22, which all cloud heights in the plots show. The curves of ch_fresco* and especially the O2-O2 CH are not as smooth as the curves of the ROCINN cloud heights and show more "oscillations". This is what is meant by steps. In Line 594 "(small oscillations)" was added to clarify "steps".

Line 612 - 630: I do not see the value in comparing the not-matched data: the sampling effects (meaning, different algorithms will include different cloud populations) would be so strong that the comparison is not meaningful. If the authors wish to retain these plots I recommend reducing or removing the analysis discussion.

Our thought here was that users are not aware of these differences if they only use one product or the products independently. Then they use the individual cloud products as they are, or only with a quality flag, and thus different cloud populations. This only becomes clear when comparing the cloud products, as we have done in these not-matched data plots.

Line 699 - 703: These conclusions are not convincing without more detailed analysis. If such additional analysis is 'out-of-scope', then at minimum, the authors should modify the text since these conclusions in current form are very speculative.

This conclusion refers to partially cloudy pixels, not fully cloudy and cloud-free pixels (as in your previous comment and which statement we took out). Since we excluded the regional across-track plots, we think this does not focus attention too much and we can leave this conclusion in the manuscript.

If more analysis can be done, here is other literature on this topic that should be cited (e.g. Maddux, B. C., Ackerman, S. A., & Platnick, S. (2010). Viewing Geometry Dependencies in MODIS Cloud Products, Journal of Atmospheric and Oceanic Technology, 27(9), 1519-1528). There are other mechanisms which could imprint a cross track variation in the CF, for example increased optical path at the scan edges.
It would also help to stratify the CF data by CH to refine the analysis: is the increase in CF at the edges similar for low-level and high-level clouds?

Further analysis in this direction is not possible for statistical reasons. In addition, the number of figures in the manuscript is already large and the attention on these plots has been reduced because we have removed the regional across-track plots.